# Bidirectional interactions between indomethacin and the murine intestinal microbiota

Xue Liang[1], Kyle Bittinger[2], Xuanwen Li[1], Darrell R Abernethy[3], Frederic D Bushman[2], Garret A FitzGerald[1]*

[1]Department of Systems Pharmacology and Translational Therapeutics, Perelman School of Medicine, University of Pennsylvania, Philadelphia, United States; [2]Department of Microbiology, Perelman School of Medicine, University of Pennsylvania, Philadelphia, United States; [3]Office of Clinical Pharmacology, Food and Drug Administration, Silver Spring, United States

**Abstract** The vertebrate gut microbiota have been implicated in the metabolism of xenobiotic compounds, motivating studies of microbe-driven metabolism of clinically important drugs. Here, we studied interactions between the microbiota and indomethacin, a nonsteroidal anti-inflammatory drug (NSAID) that inhibits cyclooxygenases (COX) -1 and -2. Indomethacin was tested in both acute and chronic exposure models in mice at clinically relevant doses, which suppressed production of COX-1- and COX-2-derived prostaglandins and caused small intestinal (SI) damage. Deep sequencing analysis showed that indomethacin exposure was associated with alterations in the structure of the intestinal microbiota in both dosing models. Perturbation of the intestinal microbiome by antibiotic treatment altered indomethacin pharmacokinetics and pharmacodynamics, which is probably the result of reduced bacterial $\beta$-glucuronidase activity. Humans show considerable inter-individual differences in their microbiota and their responses to indomethacin — thus, the drug-microbe interactions described here provide candidate mediators of individualized drug responses.

*For correspondence: garret@ upenn.edu

## Introduction

The composition of the intestinal microbiota is relatively stable in adult humans, but the taxa present differ considerably among individuals (*Gill et al., 2006*; *Arumugam et al., 2011*; *Wu et al., 2011*). The gut microbiota is influenced by host genetics (*Turnbaugh et al., 2009*; *Benson et al., 2010*), ageing (*Biagi et al., 2010*; *Agans et al., 2011*), the use of antibiotics (*De La Cochetière et al., 2005*; *Dethlefsen et al., 2008*; *Jernberg et al., 2010*; *Dollive et al., 2013*), lifestyle (*Annalisa et al., 2014*), diet (*Wu et al., 2011*; *Zoetendal and de Vos, 2014*; *Wu et al., 2014*), time of day (*Thaiss et al., 2014*; *Zarrinpar et al., 2014*; *Liang et al., 2015*), pet ownership (*Song et al., 2013*) and concomitant disease (*Zhao, 2013*; *Wu et al., 2013*). Bacterial communities in the intestine help maintain mucosal structure (*Stappenbeck et al., 2002*; *McDermott and Huffnagle, 2014*), defend against pathogens (*Littman and Pamer, 2011*), and metabolize dietary constituents such as fiber (*Sekirov et al., 2010*), peptides, proteins, (*Farthing, 2004*) and xenobiotics (*Zheng et al., 2011*; *Nicholson et al., 2012*; *Haiser et al., 2013*).

The intestinal microbiota contain ~3.3 million microbial genes, including genes encoding xenobiotics biodegradation and metabolism pathways (*Qin et al., 2010*). These bacteria are implicated in biotransformation of over 30 approved drugs by direct or indirect mechanisms (*Okuda et al., 1998*; *Sousa et al., 2008*; *Clayton et al., 2009*; *Haiser et al., 2013*). For example, bacterially generated p-

**eLife digest** The bacteria that inhabit the digestive tract do more than just help to break down food. Scientists are increasingly discovering that having a healthy and diverse community of gut bacteria is essential to overall health. Changes in these communities may increase the likelihood of harmful inflammation and diseases like obesity.

Medications may alter the gut bacteria; for example, antibiotics used to treat infections may wipe out beneficial bacteria in the digestive tract. Other drugs like nonsteroidal anti-inflammatory drugs (NSAID), which are sold over-the-counter to treat headaches and by prescription to treat pain, are known to damage the lining of the digestive tract. However, it was not clear how this affects the communities of bacteria in the gut.

Liang et al. have now used genome-sequencing tools to determine which types and quantities of bacteria are normally present at various points along the digestive tract of healthy mice. The mice were then treated with an NSAID called indomethacin, which shifted the composition of intestinal bacteria towards a pro-inflammatory structure.

Liang et al. then treated mice with antibiotics before giving them indomethacin to reduce the overall number of gut bacteria. These bacteria-depleted mice showed altered metabolism of indomethacin, and reduced blood levels of the drug, as evident in the production of fatty molecules called prostaglandins in the body. This is because the interactions between the gut bacteria and indomethacin modifies the inhibitory effect of indomethacin on enzymes known as COX-1 and COX-2. This may explain why some drugs work better in some people than others, as different people have different bacteria in their guts.

In the future, Liang et al. aim to investigate whether the communities of gut bacteria would be differently influenced by specifically inhibiting the action of either COX-1 or COX-2, given that drugs that inhibit COX-2 cause fewer gastrointestinal complications. There are also plans to explore whether alterations in the communities of gut bacteria are a driver or a passenger in gastrointestinal ailments, following ingestion of indomethacin. Previous work has shown the influence of the host molecular clock on gut bacteria. Therefore, Liang et al. will also ask if taking indomethacin at different times of day might improve how well the drug works and cause fewer side effects in animal models and eventually in humans.

---

cresol competes with the widely used analgesic acetaminophen for O-sulfonation (*Clayton et al., 2009*), and digoxin is directly inactivated by the gut Actinobacterium *Eggerthella lenta* (*Haiser et al., 2013*; *2014*).

Nonsteroidal anti-inflammatory drugs (NSAIDs) suppress prostanoid production by inhibiting the cyclooxygenase (COX)-1 and -2 enzymes. NSAIDs are widely used for relief of pain and inflammation (*Green, 2001*). A limitation of these drugs is their association with adverse gastrointestinal (GI) complications (*Graumlich, 2001*). Coincidental disruption of both COX enzymes, such as is achieved at therapeutic doses of indomethacin in humans, is necessary to evoke GI lesions in experimental animals (*Brodie et al., 1970*; *Stewart et al., 1980*). However, germ-free (*Robert and Asano, 1977*) and antibiotic-treated (*Koga et al., 1999*) rats are resistant to indomethacin-induced intestinal lesions, suggesting a role for the microbiota. Limited information is available as to the impact of NSAIDs on the composition of microbiome: indomethacin is reported to increase intestinal *Enterococcus faecalis* and decrease segmented filamentous bacteria (SFB) (*Dalby et al., 2006*), while DuP 697, a COX-2 inhibitor, increases the abundance of Gram-negative rods in rats (*Kinouchi et al., 1998*). However, whether indomethacin induces compositional changes in intestinal microbiota and whether these changes are involved in indomethacin enteropathy remains unknown. Here, we investigate interactions between indomethacin and the intestinal microbiota. Deep sequencing of longitudinal samples provided evidence that indomethacin affects the composition of the gut microbiota following both acute and chronic exposure.

Indomethacin undergoes enterohepatic recirculation (*Harman et al., 1964*) — it is glucuronidated in the liver by UDP-glucuronosyltransferases (UGTs) and the glucuronide is delivered to the SI with bile acids where it is de-conjugated and reabsorbed. Previously, a specific inhibitor of bacterial β-

glucuronidase was reported to reduce GI damage inflicted by the anticancer drug CPT-11 (*Roberts et al., 2013*) and by several NSAIDs, including diclofenac, indomethacin and ketoprofen (*Saitta et al., 2014*). Here, we provide direct pharmacokinetic evidence documenting the influence of the intestinal microbiota on indomethacin metabolism via de-glucuronidation of its metabolites during enterohepatic recirculation. Given that inter- and intra-individual variation in the intestinal microbiota is high in humans, these results suggest a possible role for the intestinal microbiota in diversification of human responses to NSAIDs.

## Results

### Geographic heterogeneity in the composition of the murine intestinal microbiota

To assess the effect of indomethacin on the intestinal microbiota, we first analyzed the composition of the luminal and tissue-associated microbiota in mice prior to drug exposure (*Figure 1*). Eight anatomical sites were analyzed — the small and large intestines were analyzed at proximal, middle, and distal sites, and cecum and feces were also compared. For each of the intestinal sites, luminal contents and mucosa were compared. We purified DNA from tissue or feces and used 16S rRNA gene sequencing and community analysis implemented using the QIIME pipeline (*Caporaso et al., 2010b*) to characterize geographic differences. The microbiota were compared between GI sites using UniFrac (*Lozupone and Knight, 2005*), and visualized using Principal Coordinate Analyses (PCoA) (*Caporaso et al., 2010b*) of unweighted UniFrac distances (describing the bacterial lineages presented in samples) and weighted UniFrac distances (describing the proportions of bacterial lineages in samples).

The composition of the intestinal microbiota varied considerably by anatomical site (*Figure 1A*). Most samples were dominated by Bacteriodetes and Firmcutes. In the lumen, the cecum was dominated by Firmicutes (58.64% ± 3.49%), while Bacteroidetes were more abundant in the large intestine (LI) (50.51% ± 4.03% at proximal, 67.91% ± 2.72% at middle, 72.14% ± 2.25% at distal LI). (*Figure 1B*, upper). In the mucosa, Firmicutes were dominant in SI (52.40% ± 6.56% at proximal, 48.34% ± 5.65% at middle, 45.49% ± 6.61% at distal SI), cecum (60.48% 2.76%), and proximal LI (85.70% ± 2.00%) (*Figure 1B*, lower). Proteobacteria were abundant in the proximal SI (14.0% ± 4.27% in the lumen, 22.31% ± 6.37% in the mucosa) and distal LI (11.34% ± 2.72%), though results were more heterogeneous than at other sites, probably in part reflecting low bacterial biomass in the starting material (*Figure 2A*). LI lumen and fecal compositions exhibited considerable similarity (*Figure 1A*, p<0.001 for both weighted and unweighted UniFrac distance, ADONIS test). The composition of the microbiota in the luminal content differed from that at the mucosal surface throughout the intestine (p<0.001 for both weighted and unweighted UniFrac distance, ADONIS test), as indicated by the separation of luminal content samples and mucosal tissue samples (*Figure 1A*).

### Acute indomethacin exposure induces compositional changes in intestinal microbiota

A single dose of 10 mg/kg indomethacin was introduced into mice by gavage to test for effects on the microbiota. This acute dose is clinically relevant and is thus widely used in animal models (*Fukumoto et al., 2011*; *Tanigawa et al., 2013*). Mucosal erosion and ulcerations were observed in SI 24 hr after indomethacin treatment (*Figure 2—figure supplement 1*) but not with vehicle control (PEG400). To analyze the compositional changes in microbial communities before the appearance of indomethacin-induced lesions, we analyzed animals after 6-hr of treatment. Urinary prostanoid metabolites (*Figure 2—figure supplement 2*) were suppressed, predominantly reflective of COX-1 (PGD-M and Tx-M) and COX-2 (PGI-M and PGE-M) inhibition. Indomethacin was detected in plasma and urine as well as in luminal contents and mucosal tissues along the intestine (*Figure 2—figure supplement 3*), suggesting local and systemic exposure to the drug.

Sixty mice were analyzed to characterize microbial responses to indomethacin exposure. Mice were randomly divided into three equal groups, receiving (i) 10 mg/kg indomethacin in PEG400 vehicle; (ii) PEG400 only and (iii) an untreated group. Examination of the bacterial biomasses using 16S rRNA gene qPCR revealed no effect of indomethacin, although a vehicle effect was evident (*Figure 2A*). Mice treated with PEG400 showed a decrease in luminal biomasses and an increase in

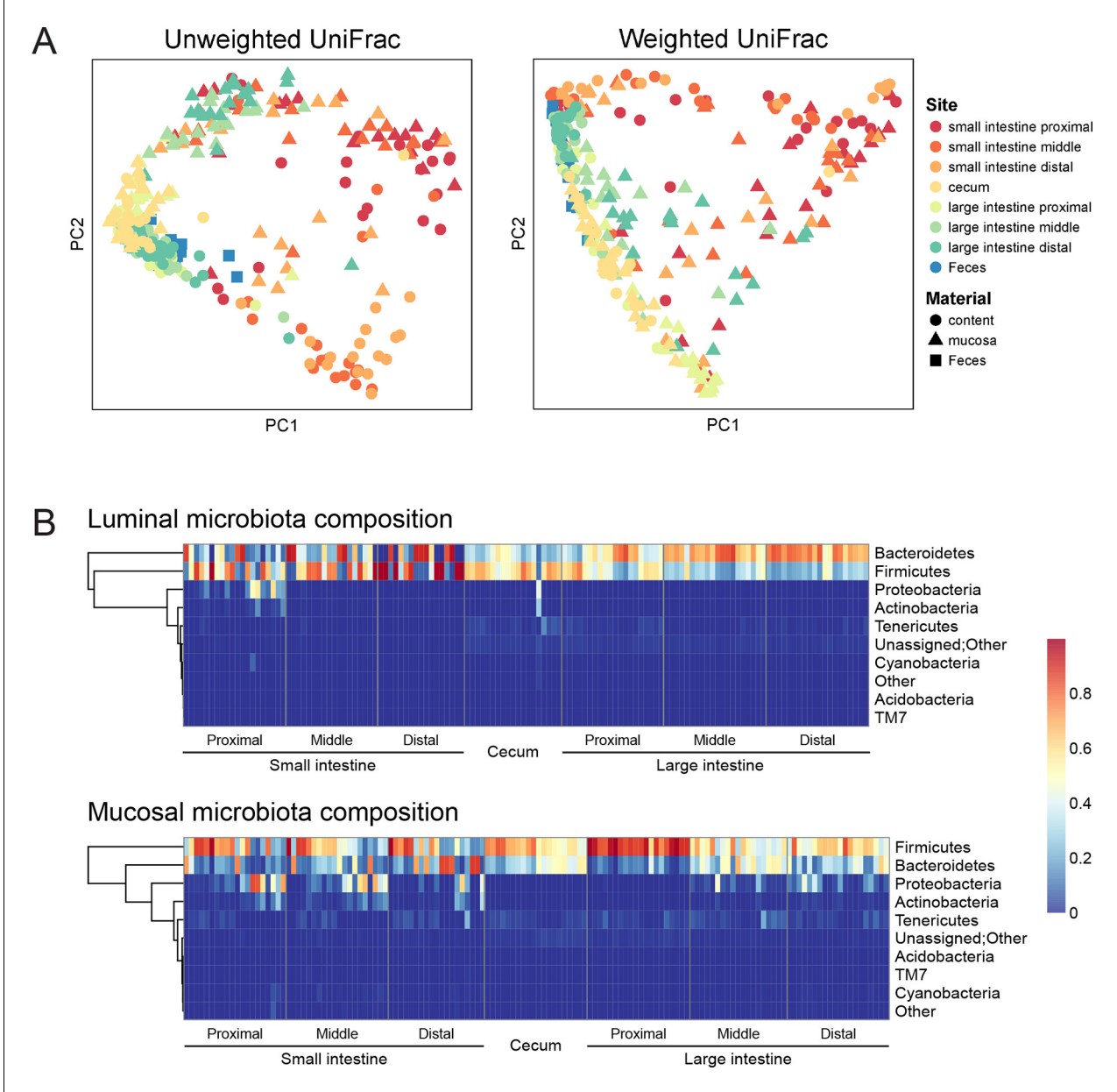

**Figure 1.** Geographic heterogeneity of basal intestinal microbiota composition along the intestine in mice. Bacterial communities colonized in the mouse intestine were profiled using 16S rRNA gene sequencing and analyzed using QIIME (*Caporaso et al., 2010b*). (A) Principal coordinates analysis (PCoA) of unweighted (left) and weighted (right) UniFrac values (*Lozupone et al., 2011*), depicting the comparison of microbial communities from luminal content (round), mucosal tissue (triangle), or feces (square). The base line microbiota compositions along the intestine are heterogeneous at anatomical sites. Each point represents a sample, and each sample is colored according to the habitat sites in the intestine. N=17–20. (B) Heat map of the microbiota composition in luminal content (upper) and mucosal tissue (lower) along the intestine. Each column represents sample, and each row represents one phylum. The proportions of phyla are indicated by the color code to the right. Anatomical sites of the intestine are indicated at the bottom. N=17–20.

mucosal tissue biomasses in the distal end of the LI, compared to untreated mice. Similarly in humans, treatment with Golytely, which contains PEG 3350, has been associated with changes in the mucosal-associated microbiota in colon (*Harrell et al., 2012*).

Microbial community structure along the intestine was quantified for observed species, which reflects the richness by measuring the number of operational taxonomic units (OTUs) and the Shannon Index, which indicates the diversity by taking account of both the number of OTUs and the

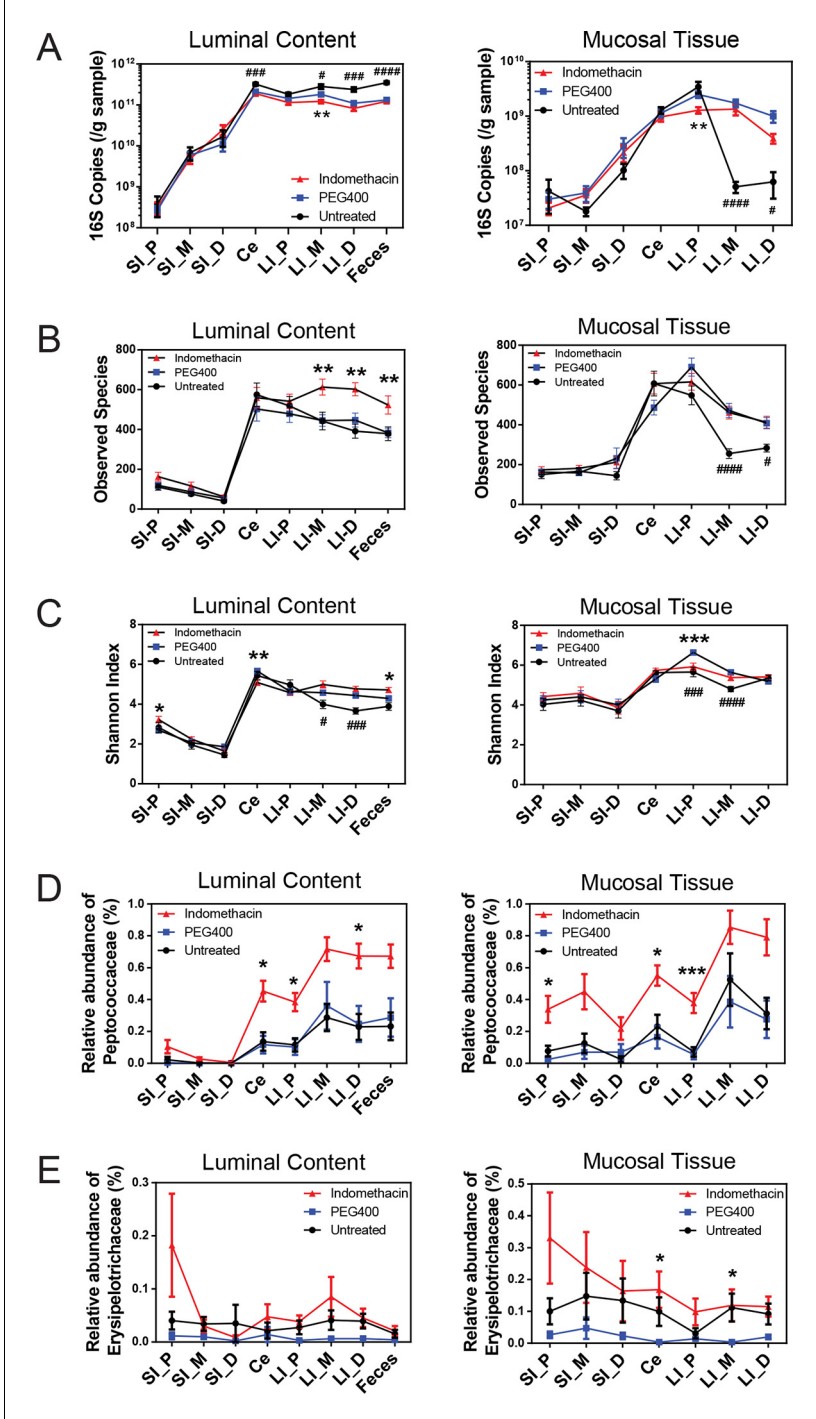

**Figure 2.** Indomethacin induces changes in microbial composition along the intestine in mice. Bacterial load in samples were inferred from 16S rRNA gene quantitative PCR. Bacterial communities colonized in the mouse intestine were profiled using 16S rRNA gene sequencing and analyzed using QIIME (*Caporaso et al., 2010b*). (**A**) 16S rRNA gene copies per gram of luminal contents (left) and mucosal tissues (right) at anatomical sites along the intestine in indomethacin (red), PEG400 (blue), and untreated (black) groups. Microbial loads at anatomical sites along the intestine are barely different between PEG400 and indomethacin groups, although PEG400 causes changes by itself. **p<0.01 by multiple t test comparing PEG400 versus indomethacin groups, FDR corrected. #p<0.05, ###p<0.001, ####p<0.0001 by multiple t test comparing untreated versus PEG400 groups, FDR corrected. N=20/group. Mean ± S.E.M. shown. SI, small intestine; Ce, cecum; LI, large intestine. P, proximal; M, middle; D, distal. Observed Species (**B**) and Shannon Index (**C**) are used to estimate richness and diversity of microbial communities in luminal content (left) and mucosal tissue (right) at anatomical sites along the intestine in indomethacin (red), PEG400 (blue), and untreated (black) groups. Indomethacin altered microbial diversity in the distal intestine, although PEG400 also causes changes in the distal intestine by itself. *p<0.05, **p<0.01, ***p<0.001 by multiple t test comparing

*Figure 2 continued on next page*

*Figure 2 continued*

PEG400 versus indomethacin groups, FDR corrected. #p<0.05, ###p<0.001, ####p<0.0001 by multiple t test comparing untreated versus PEG400 groups, FDR corrected. N=20/group. Relative abundance of *Peptococcaceae* (D) and *Erysipelotrichaceae* (E) at anatomical sites along the intestine are significantly elevated in indomethacin (red) group than in PEG400 (blue) and untreated (black) group in both luminal content and mucosal tissues of the distal gut. *p<0.05, ***p<0.001 by QIIME analysis, FDR corrected. Mean ± S.E.M. shown. SI, small intestine; Ce, cecum; LI, large intestine. P, proximal; M, middle; D, distal.

The following figure supplements are available for figure 2:

**Figure supplement 1.** Indomethacin induces small intestinal damage in C57BL/6 mice.

**Figure supplement 2.** Inhibitory effects of acute indomethacin treatment on COX-1 and COX-2 in C57BL/6 mice.

**Figure supplement 3.** C57BL/6 mice are systemically and locally exposed to indomethacin.

**Figure supplement 4.** Inhibitory effects of chronic indomethacin treatment on COX-1 and COX-2 in C57BL/6 mice.

**Figure supplement 5.** Chronic indomethacin treatment induces changes in microbial composition along the intestine in mice.

evenness of distribution of reads among the OTU categories. Comparison between the indomethacin and PEG400 groups revealed changes primarily in the LI. Indomethacin caused an increase in richness (*Figure 2B*) in the middle (p<0.01, FDR corrected) and distal LI luminal content (p<0.01, FDR corrected), as well as in feces (p<0.01, FDR corrected), without influencing the mucosal tissues. Diversity (*Figure 2C*) was decreased in the luminal content of the cecum (p<0.01, FDR corrected) and in the mucosal tissue of proximal LI (p<0.001, FDR corrected), while it was increased in feces (p<0.05, FDR corrected). PEG400 alone increased microbial diversity in the distal LI.

The abundance of some bacterial lineages was also affected by indomethacin. *Peptococcaceae* expanded in the luminal content of cecum (0.45% ± 0.07%), the proximal LI (0.38% ± 0.06%) and distal LI (0.67% ± 0.06%), as well as in mucosal tissues of the cecum (0.55% ± 0.06%) and the proximal LI (0.38% ± 0.06%) (*Figure 2D*). *Erysipelotrichaceae* expanded in the mucosal tissues of cecum (0.17% ± 0.06%) and middle LI (0.12% ± 0.05%), yet were less affected in the luminal content (*Figure 2E*). Separation of bacterial communities between indomethacin- or PEG400-treated mice was most evident in the LI mucosal tissues (p=0.004 for proximal LI, p=0.009 for middle LI, p=0.009 for distal LI; weighted UniFrac distance).

## Chronic indomethacin exposure also induces compositional changes in the intestinal microbiota

Since indomethacin is also chronically used in humans, we sought to understand the effects of chronic drug exposure in the mouse model. We thus introduced indomethacin in the diet, bypassing vehicle effects, in a second age and gender matched mouse cohort. Twenty mice were randomly divided into two groups, receiving control diet or an indomethacin-supplemented diet (20 ppm), administered for 7 days, and were then sacrificed one day later. This dose was selected based on the previous work (*Chiu et al., 2000*; *Fjære et al., 2014*; *Leibowitz et al., 2014*) and to ensure tolerability. Indomethacin significantly suppressed urinary prostanoid metabolites (*Figure 2—figure supplement 4*), suggesting COX-1 (PGD-M and Tx-M) and COX-2 (PGI-M and PGE-M) inhibition. The suppression was to a similar extent as was observed for the 6-hr treatment (*Figure 2—figure supplement 2*). Chronic indomethacin treatment was associated with a decrease in the Shannon Index in the luminal content of cecum. Richness and diversity were not affected at other GI sites (*Figure 2—figure supplement 5A,B*).

Several compositional changes detected in the acute treatment study were reproduced after chronic treatment. *Peptococcaceae* increased in relative abundance as in the acute study – in the chronic study this lineage expanded in the luminal content of cecum (0.28% ± 0.08%), proximal LI (0.35% ± 0.09%) and distal LI (0.43% ± 0.13%), as well as in mucosal tissues of cecum (0.10% ± 0.02%) and proximal LI (0.11% ± 0.02%) (*Figure 2—figure supplement 5C*). *Erysipelotrichaceae*,

which also expanded in the acute treatment, expanded in the chronic treatment in mucosal tissues of cecum (0.21% ± 0.07%) (*Figure 2—figure supplement 5D*).

Thus, both acute and chronic dosing affected the microbiota. A single oral dose of indomethacin induced alterations in microbial diversity in the distal intestine and caused compositional changes along the intestine, with only slight effects on microbial biomasses. Chronic indomethacin treatment exhibited some of the same effects on microbial composition for both the lineages affected and directions of change.

## Indomethacin induces longitudinal compositional changes in the fecal microbiota

Collection of fecal pellets from the same mouse before and after indomethacin treatment allowed analysis of within-individual compositional changes over time (*Figure 3A*). We detected significant clustering between 0 hr and 6 hr microbial communities in the indomethacin-treated group (p<0.01, ADONIS test) and PEG400 group (p<0.05, ADONIS test), but not in untreated group (p>0.5, ADONIS test). However, drug treatment explains more of the observed variation in the indomethacin group ($R^2$ = 0.22, ADONIS test) than in PEG400 group ($R^2$ = 0.05, ADONIS test), indicating indomethacin had a greater effect in modulating fecal microbiota composition than PEG400. The influence of indomethacin was not due to changes in microbial biomasses, since there were no significant changes in 16S rRNA gene copy number between PEG400 and indomethacin groups, as measured by qPCR (*Figure 3B*). There may be a vehicle-induced decrease in microbial biomass, likely due to its purgative effect. Indomethacin also induced an increase in the Shannon Index (*Figure 3C*, right), without influencing observed species in the fecal microbiota, suggestive of an increase in evenness associated with drug exposure.

A phylum-level shift was evident in indomethacin-treated mice (*Figure 3D*), with significantly decreased Bacteroidetes (73.84% ± 3.39% at 0 hr versus 56.02% ± 2.04% at 6 hr, p<0.01 after FDR correction) and increased Firmicutes (24.95% ± 3.25% at 0 hr versus 41.97% ± 2.01% at 6 hr, p<0.01 after FDR correction). These trends were also detectable at lower taxonomic levels (*Figure 3E*), including decreased *S24-7 spp.* (Bacteroidetes), and increased *Ruminococcus*, *Lachnospiraceae*, and *rc4-4* (Firmicutes), and *Anaeroplasma* (Tenericutes).

PEG400 treatment increased *Clostridiales spp.* and *Lactobacillus*, while decreased *Ruminococcaceae spp.* and *Oscillospira* (*Figure 3—figure supplement 1*), consistent with the clustering observed in the PCoA plots (*Figure 3A*). However, indomethacin may counteract the effect of PEG400, leading to the increase of *Ruminococcaceae spp.*, or unchanged *Lactobacillus* and *Oscillospira*.

The chronic indomethacin treatment introduced in the diet had diverse effects on the fecal microbiota. Richness, diversity, and the relative abundance of Bacteroidetes and Firmicutes were not significantly affected (*Figure 3—figure supplement 2A,B*). However, changes at the genus level including expansion of *Ruminococcus* (0.49% ± 0.09% on day 0 versus 0.80% ± 0.13% on day 8, p<0.05) and *Anearoplasma* (0.24% ± 0.05% on day 0 versus 0.72% ± 0.20% on day 8, p<0.01) (*Figure 3—figure supplement 3*), matching effects seen in the acute dosing study.

## Indomethacin metabolism is altered in microbiota-perturbed mice

To investigate the impact of intestinal microbiota on the metabolism of indomethacin, we used antibiotics to deplete the microbiota, then compared metabolism in treated and control mice. Mice were treated with either control (water) or an antibiotic cocktail (1 g/L neomycin and 0.5 g/L vancomycin) for 5 days. Fecal pellets were collected daily. The 16S rRNA gene copy numbers in feces were reduced by five orders of magnitude after antibiotic treatment, and this was maintained for up to 2 days after treatment cessation (*Figure 4A*). Body weight, food intake, and water intake were not affected by antibiotic treatment over the time-course studied (*Figure 4—figure supplement 1A–C*). Microbial diversity analysis revealed a significant decrease, starting at day 4, with recovery still incomplete by day 7 (*Figure 4B*). After 5 days of antibiotic treatment, mice showed a significantly shifted composition of the fecal microbiota, with a reduction in Bacteroidetes and Firmicutes, and a concomitant expansion of Proteobacteria (*Figure 4—figure supplement 1D*).

Mice treated with the antibiotic cocktail or control were administered 10 mg/kg indomethacin by gavage on day 5 followed by sequential blood sampling over 48 hr. In the antibiotic-treated mice, the oral clearance of indomethacin was increased by 19.6% (*Figure 4C*), and the elimination rate

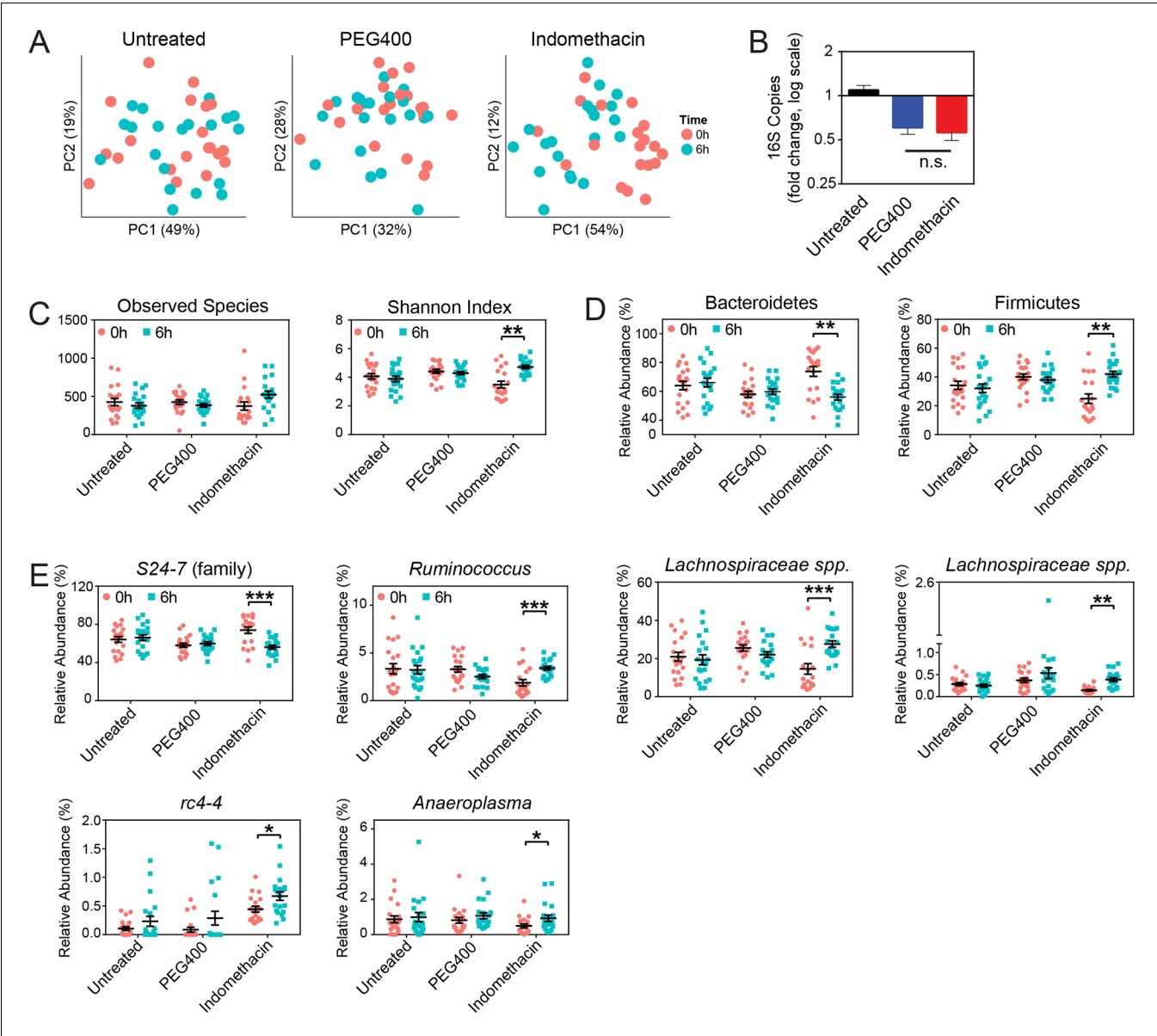

**Figure 3.** Indomethacin induces longitudinal changes in fecal microbiota composition. Microbiota composition in fecal pellets before (0 hr) and after (6 hr) treatment with or without indomethacin or PEG400 is analyzed by 16S rRNA gene profiling, including sequencing and quantitative PCR. (**A**) Principal coordinates analysis (PCoA) of weighted UniFrac values (*Lozupone et al., 2011*), comparing the fecal microbial communities at 0 hr (black) versus 6 hr (blue) of untreated (left), PEG400 (middle), and indomethacin (right) groups. Each point represents a sample. Fecal microbial communities at 0 hr and 6 hr are not separated in untreated group (p>0.5), and significantly clustered in PEG400 group (p<0.5) and in indomethacin group (p<0.01). Clustering was analyzed by ADONIS test. (**B**) 16S rRNA gene copies per gram of feces at 0 hr and 6 hr (left), and Fold changes (right) in indomethacin (red), PEG400 (blue), and untreated (black) groups. Both PEG400 and indomethacin groups have lower bacterial loads at 6 hr, whereas these are no between-group differences at 0 hr or 6 hr. ****p<0.0001 by Mann-Whitney test comparing 0 hr versus 6 hr. N=20/group. Mean ± S.E.M. shown. (**C**) Both Observed Species (left) and Shannon Index (right) are increased at 6 hr in indomethacin-treated mice, while unchanged in Untreated and PEG400 groups. **p<0.01 by multiple t test, FDR corrected. N=19–20/group. Mean ± S.E.M. shown. (**D**) The relative abundance of Bacteroidetes (left) is decreased and that of Firmicutes (right) is increased at 6 hr (blue) after indomethacin treatment. **p<0.01 by multiple t test, FDR corrected. N=19–20/group. (**E**) Indomethacin induced a decrease in the relative abundance of *S24-7* (family), and increases in those of *Ruminococcus, Lachnospiraceae sp., Lachnospiraceae sp., rc4-4*, and *Anaeroplasma* at 6 hr (blue). *p<0.05, **p<0.01, ***p<0.001 by QIIME analysis, FDR corrected. N=19–20/group. Mean ± S.E.M. shown.

The following figure supplements are available for figure 3:

**Figure supplement 1.** Longitudinal effects of acute indomethacin treatment in fecal microbiota composition.

*Figure 3 continued on next page*

*Figure 3 continued*

**Figure supplement 2.** Longitudinal effects of chronic indomethacin treatment in fecal microbiota composition.

**Figure supplement 3.** Longitudinal effects of chronic indomethacin treatment on genera abundance in fecal microbiota.

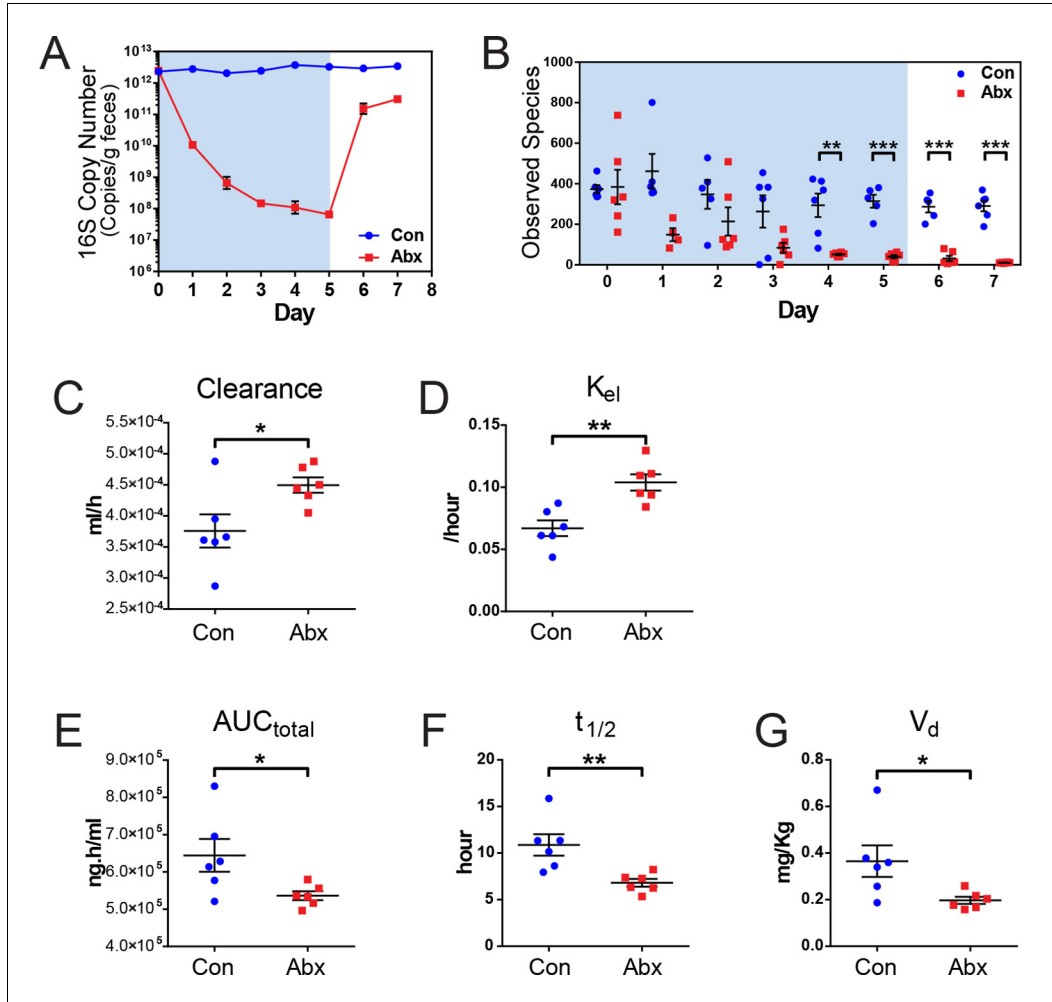

**Figure 4.** Microbiota-depletion with antibiotics alters the pharmacokinetics of indomethacin in mice. Mice were subjected to control water (Con) or antibiotic cocktail (Abx, neomycin and vancomycin) for 5 days (blue-shaded area). Upon the cessation of 5-day treatment, mice were administered by gavage with 10 mg/kg indomethacin. Plasma was collected sequentially for pharmacokinetic analysis. Fecal microbiota compositions over time were analyzed using 16S rRNA gene profiling. (A) Longitudinal analysis of 16S rRNA gene copies per gram of feces reveals a significant reduction in microbial load in Abx group (red). (B) Longitudinal analysis of observed species reveals decreased microbial richness in Abx group (red). **$p < 0.01$, ***$p < 0.001$ by multiple t test, FDR corrected. N=4–6/group. Mean ± S.E.M. shown. In antibiotic-treated mice (red), indomethacin has increased oral clearance (C) and elimination rate constant ($K_{el}$) (D), as well as decreased area under the curve ($AUC_{total}$) (E), half-life ($t_{1/2}$) (F), and apparent volume of distribution ($V_d$) (G). *$p < 0.05$, **$p < 0.01$ by Mann-Whitney test. N=6/group. Mean ± S.E.M. shown.

The following figure supplement is available for figure 4:

**Figure supplement 1.** Antibiotic-treatment causes compositional changes in intestinal microbiota without affecting body weight, food intake, and water intake in C57BL/6 mice.

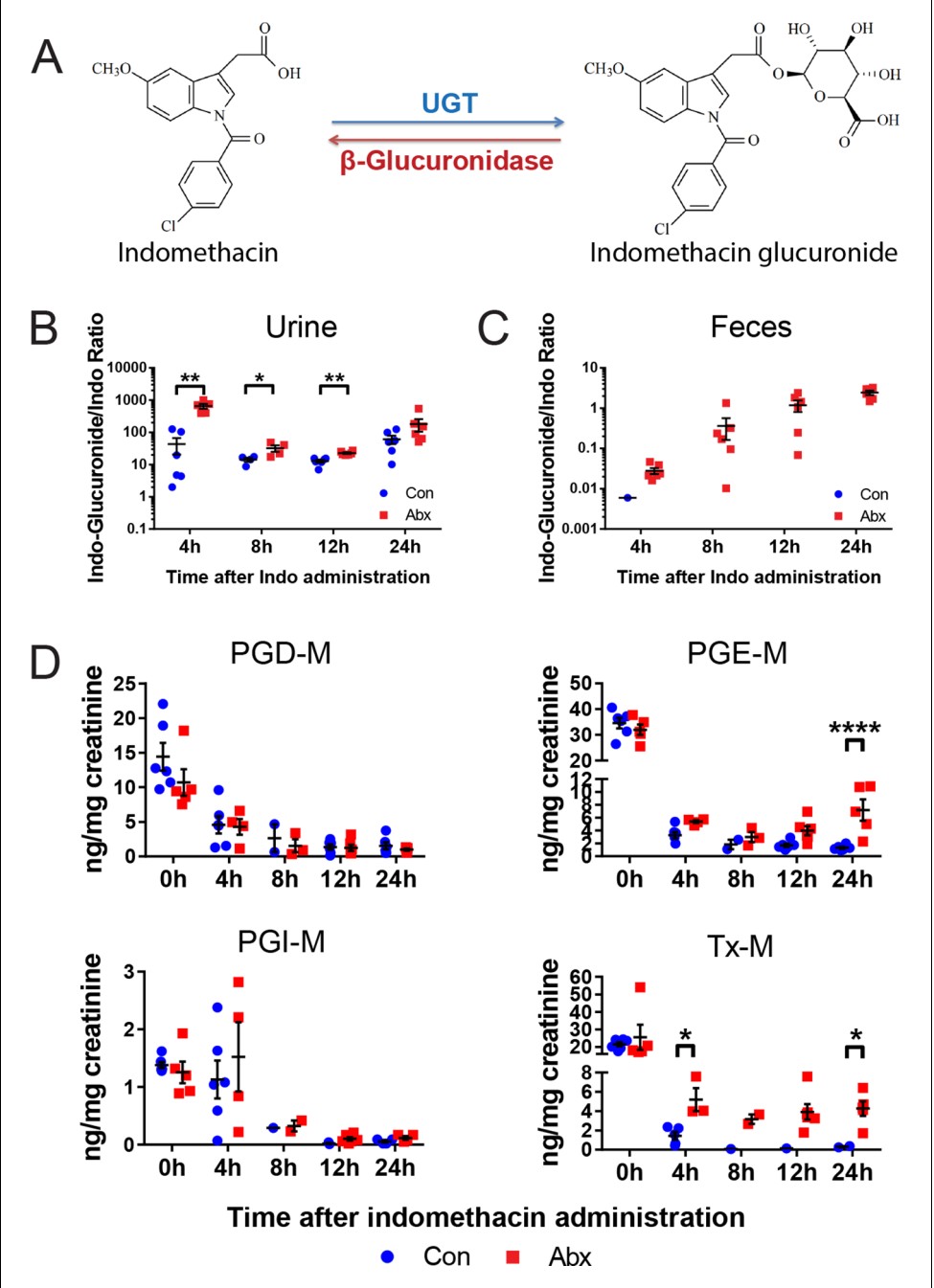

**Figure 5.** Metabolism and efficacy of indomethacin in antibiotic-treated mice are altered. Upon the cessation of 5-day treatment with antibiotic cocktail (Abx, neomycin and vancomycin) or control water (Con), mice were administered by gavage with 10 mg/kg indomethacin. Urine and feces were collected at indicated time for metabolic analysis. (**A**) Chemical structures of indomethacin (left) and indomethacin glucuronide (right). Enzyme catalyzing the glucuronidation is UDP-glucuronosyltransferase (UGT), and the one catalyzing the de-glucuronidation is β-glucuronidase. The ratio of indomethacin-glucuronide to indomethacin in urine (**B**) and feces (**C**) are higher in Abx group (red) than in Con group (blue). *$p<0.05$, **$p<0.01$ by Mann-Whitney test. N=6/group. Mean ± S.E.M. shown. (**D**) Urinary prostanoid metabolites were analyzed with LC/MS and values are corrected by creatinine. In Con mice (blue), all metabolites were reduced time-dependently. In Abx mice (red) PGD-M and PGI-M remained suppressed 24 hr after indomethacin, whereas PGE-M and Tx-M concentrations recovered more quickly. Two-way ANOVA revealed significant effect of time in PGD-M ($p=0.001$) and PGI-M ($p=0.0004$), and significant antibiotic effect of PGE-M ($p<0.0001$) and Tx-M ($p=0.0002$).In Abx mice, PGE-M was higher mice at
*Figure 5 continued on next page*

*Figure 5 continued*

24 hr, and Tx-M was higher at 4 hr and 24 hr. N=6/group. *p<0.05, **p<0.01 by multiple comparison test, adjusted. Mean ± S.E.M. shown.

The following figure supplement is available for figure 5:

**Figure supplement 1.** *β*-glucuronidase catalyzes de-glucuronidation reaction.

constant $K_{el}$ was increased by 55.2% (*Figure 4D*), indicating an increased elimination of indomethacin. The total area-under-the-curve ($AUC_{total}$) of indomethacin, which is a measurement of total drug exposure, was decreased by 16.8% in antibiotic-treated mice (*Figure 4E*). The half-life ($t_{1/2}$) of indomethacin was decreased by 37.5% (*Figure 4F*), and the apparent volume of distribution ($V_d$) of indomethacin was decreased by 46.1% (*Figure 4G*) in antibiotic-treated mice. The variances of $AUC_{total}$, $t_{1/2}$ and $V_d$ of indomethacin were significantly smaller in antibiotic-treated mice than in control mice (p=0.01 for $AUC_{total}$, p=0.04 for $t_{1/2}$, and p=0.006 for $V_d$, F test), suggesting intestinal bacteria as one of the sources of inter-mouse variation in response to indomethacin.

A second set of age and gender matched mice treated with or without the antibiotic cocktail were administered 10 mg/kg indomethacin by gavage on day 5, and urine and feces were collected for the following 24 hr. Detection of indomethacin and indomethacin-glucuronide was confirmed by incubating samples with or without β-glucuronidase. As shown in the representative spectra (*Figure 5—figure supplement 1A*), after incubation with β-glucuronidase, the peak of indomethacin glucuronide was diminished and that of indomethacin greatly increased. This change was detected in each of the samples studied (*Figure 5—figure supplement 1B,C*). To evaluate enzyme activity, we compared the ratio of indomethacin-glucuronide to indomethacin in mice pretreated with or without the antibiotic cocktail. In urine, the ratio was significantly higher in antibiotic-treated mice for the first 12 hr following indomethacin administration (93.3% higher at 4 hr, 55.6% higher at 8 hr, 43.4% at 12 hr; *Figure 5B*). In feces, indomethacin-glucuronide was barely detectable in control mice, yet was readily detected in antibiotic-treated mice (*Figure 5C*). Indomethacin suppressed urinary prostanoid metabolites irrespective of treatment with the antibiotic cocktail (*Figure 5D*). In control mice, indomethacin reduced these metabolites in a time-dependent fashion. A similar pattern was evident for PGD-M and PGI-M in antibiotic-treated mice. However, concentrations of the most abundant prostanoids, PGE-M and Tx-M, were suppressed to a lesser degree by indomethacin, and their concentration started to recover faster in antibiotic-treated mice compared to control mice. Evidently β-glucuronidase-catalyzed de-glucuronidation was impaired due to antibiotic-treatment, partially suppressing the inhibitory effect of indomethacin on COX enzymes. Thus, the intestinal microbiota influences the disposition and efficacy of indomethacin in the host, at least in part by regulating its de-glucuronidation and reabsorption from the intestine.

## Discussion

Here, we investigated interactions between the intestinal microbiota and the NSAID indomethacin. We documented a bidirectional interaction — indomethacin altered the composition of the intestinal microbiota, and the intestinal microbiota altered indomethacin metabolism. The presence of intestinal bacteria boosted the circulating concentrations of indomethacin, which resulted in measurable changes in prostaglandin metabolism. Apparently, bacterial-encoded de-gluconuridation enzymes deconjugated the indomethacin-gluconuride catabolic intermediate to allow indomethacin recycling.

We showed that only a single oral dose of indomethacin was sufficient to perturb the intestinal microbiota, specifically within the cecum, LI and feces. Drug-induced effects were less evident in the SI, possibly attributable to higher inter-individual variance in composition (*Figure 1A*) which limits the ability to detect the impact of interventions. The selected dose of indomethacin resulted in systemic drug exposure, including detection in luminal contents, along the mucosa of the SI and LI, and in plasma, urine and feces. Reflective of its mechanism of analgesic and anti-inflammatory action, this dose of indomethacin suppressed endogenous biosynthesis of prostaglandins derived primarily from COX-1 ($PGD_2$ and Tx) and COX-2 ($PGI_2$ and $PGE_2$) (*McAdam et al., 1999*; *Ricciotti and Fitz-Gerald, 2011*), as reflected by urinary excretion of their major metabolites. It also resulted in

intestinal damage, reminiscent of the upper and lower GI complications induced by NSAIDs in humans (*Allison et al., 1992*; *Smale et al., 2001*; *Sostres et al., 2013*).

The alterations in the intestinal microbiota induced by indomethacin — specifically expansion of pro-inflammatory bacteria — may have functional consequences. For example, indomethacin induces expansion of *Erysipelotrichaceae* in LI mucosa. This Gram-positive family of Firmicutes has been associated with parenteral nutrition-induced liver injury (*Harris et al., 2014*), obesity (*Zhang et al., 2009*), colorectal cancer (*Zhu et al., 2014*; *Chen et al., 2012*), and Crohn's disease (*Kaakoush et al., 2015*). Indomethacin also induced pro-inflammatory shifts in the composition of fecal microbiota, for example, a significantly increased ratio of Firmicutes to Bacteroidetes. This shift has been previously reported in genetically obese mice (*Ley et al., 2005*; *Turnbaugh et al., 2006*), obese children (*Bervoets et al., 2013*), and obese adults (*Ley et al., 2006*). A decrease of *S24-7*, a family of Bacteroidetes, such as induced here by indomethacin, has been observed in a mouse model of colorectal cancer (*Liang et al., 2014*) and in mice with high fat diet-induced obesity (*Evans et al., 2014*). *Lachnospiraceae*, also increased following indomethacin administration, have been associated with lupus (*Zhang et al., 2014*), drug-induced liver toxicity (*Xu et al., 2015*) and the development of obesity and diabetes in genetically susceptible mice (*Kameyama and Itoh, 2014*). While the importance of many of these observations remains to be established, they do suggest possible mechanisms of adverse health consequences.

The observed influence of PEG400, the vehicle used in our acute dosing study, on intestinal microbiota was not surprising. PEG used in bowel preparation in humans has been reported to have an effect on microbial diversity and composition (*Harrell et al., 2012*). PEG binds water and prevents absorption in the large intestine, and so may hydrate the LI microbiome or flush upstream bacteria to downstream sites. However, our comparative study design allowed us to detect indomethacin effects independently of vehicle effects.

Despite the PEG effect, a number of compositional changes quantified in the acute study were reproduced in the chronic treatment study. Expansion of *Peptococcaceae* and *Erysipelotrichaceae* in the GI tract and *Ruminococcus* and *Anearoplasma* in feces were seen in both studies. Whether and how these strains contribute to indomethacin-induced GI toxicity is unclear and warrants further study. Several changes in the microbiota observed in the acute study were not detected in the chronic study, which could be due to their involvement in the initial response only, or due to gradual recovery of the intestinal microbiota during chronic drug exposure. Systemic exposure to indomethacin was also reduced to insure tolerability in the chronic dosing study.

The influence of indomethacin on the LI microbiota may be clinically important, since indomethacin causes complications in the lower GI tract with a similar incidence as in the upper GI tract (*Sostres et al., 2013*). The patients receiving indomethacin were reported to show increased large intestinal permeability (*Suenaert et al., 2003*), colonic ulceration and bleeding (*Oren and Ligumsky, 1994*), multiple colonic perforations (*Loh et al., 2011*), and hemorrhage (*Langman et al., 1985*). Compositional changes in LI microbiota associated with indomethacin administration may be involved in inflammatory processes directly in the lower GI tract, and might also affect the upper GI tract indirectly. For example, metabolic products produced by the LI microbiota may modify the local environment or enter the circulation, hence changing inter-bacterial interactions or host physiology at other sites including the SI.

Here, we provide pharmacokinetic evidence that indomethacin metabolism is influenced by intestinal bacteria — specifically, antibiotic suppression of intestinal bacteria significantly reduced the level of its de-glucuronidation. In the absence of bacterial de-gluconuronidation, indomethacin reabsorption into the circulation was reduced, resulting in increased elimination, a shortened half-life and reduced drug exposure. Concomitantly, indomethacin-induced suppression of $PGE_2$ and Tx was attenuated in microbiota-depleted mice, suggesting a reduction in indomethacin efficacy resulting from the loss of intestinal bacteria. The reduction in drug exposure due to antibiotic treatment may also explain the attenuated enteropathy associated with indomethacin in rats pretreated with antibiotics (*Koga et al., 1999*), and further support previous reports from Boelsterli and colleagues (*Saitta et al., 2014*), which reported that a small molecule inhibitor of bacterial β-glucuronidase were protective against NSAID-induced ulcerations in small intestine.

Indomethacin shows considerable inter-individual variation in pharmacokinetics, efficacy and risk of GI complications (*Brune, 1985*; *1987*) that is not explained by human genetic variation (*Nakajima et al., 1998*; *Martin et al., 2001*). Our finding that depletion of intestinal bacteria

significantly reduced inter-mouse variability of half-life, volume of distribution, and drug exposure suggests bacteria-mediated metabolism as a source of variation in drug response. Given that multiple human intestinal bacteria encode β-glucuronidase genes (*Dabek et al., 2008*; *Gloux et al., 2011*) and that the intestinal microbiota are variable amongst individuals (*Wu et al., 2011*; *Arumugam et al., 2011*), differences in bacteria-mediated metabolism provide a reasonable explanation for inter-individual differences in indomethacin pharmacokinetics (*Brune, 1985*; *1987*).

The pharmacokinetics of orally dosed indomethacin shows circadian variation both in humans and in rats, which may also reflect a contribution from the microbiota. A prolonged apparent half-life of indomethacin was observed in patients dosed in the evening compared to those dosed in the morning or at noon (*Guissou et al., 1983*), accompanied by fewer undesirable effects (*Levi et al., 1985*). We and others have shown that the intestinal microbial load and composition varies during the day-night cycle (*Thaiss et al., 2014*; *Zarrinpar et al., 2014*; *Liang et al., 2015*), including strains bearing β-glucuronidase activity. Taken together with our findings here, intra-individual variation of the intestinal microbiota during the 24-hr light-dark cycle may contribute to indomethacin chronopharmacology.

In summary, a single oral dose of indomethacin elicited changes in composition and diversity of the microbiota. Reciprocally, the intestinal microbiota influenced indomethacin metabolism and its effectiveness as a systemic prostaglandin inhibitor. These results suggest that a dynamic interplay with the intestinal microbiome may contribute to adverse effects and variability in efficacy of indomethacin and perhaps other drugs.

## Materials and methods

### Animals

All C57BL/6 mice were purchased from the Jackson Laboratory and housed in our animal facility for at least 2 weeks before the performance of experiments. Male mice 10–14 weeks of age were used for all experiments. All animals were fed *at libitum* with regular chow diet (5010, LabDiet, St. Louis, MO, USA) for the course of study. Mice were kept under 12-hr light/12-hr dark (LD) cycle, with lights on at 7 am and off at 7 pm. Experimental protocols were reviewed and approved by the Institute for Animal Care and Use Committee at the University of Pennsylvania.

### Study design

All chemicals used were purchased from Sigma-Aldrich (St. Louis, MO, USA) unless otherwise stated.

### Study 1: The effect of indomethacin on the microbiota composition in mouse gut

For the acute dosing study, the sample size was chosen based on the measured inter-mouse variability and the magnitude of the effect that we wished to detect (mean 25% change with $\alpha = 0.05$ and $1-\beta = 0.8$). At 12 pm, non-fasted mice were administered by gavage (i) 10 mg/kg indomethacin (in PEG400), (ii) PEG400, or (iii) left untreated. In the experiment to evaluate indomethacin-induced intestinal damage, mice were sacrificed 24 hr after administration to harvest their small intestines for future histological analysis. In the experiment to evaluate indomethacin-induced compositional changes in gut microbiota, fecal pellets were collected prior to and 6 hr post drug administration. Then, mice were sacrificed to sample the luminal contents and adjacent mucosal tissues at the proximal, middle, and distal regions of SI and LI, as well as at the tip of cecum. Briefly, GI tract was opened; luminal content was gently scraped from the top without contact with mucosa, and the adjacent area was cut out and subjected to serial gentle wash in water until nothing visible fall off. All samples were weighed, placed in empty vials, and immediately stored at -80°C for microbiota composition analysis. In another experiment to evaluate the inhibitory effect of indomethacin, mice were housed individually in metabolic cages after drug administration for 6 hr to collect their urine and fecal pellets. Then mice were sacrificed and their intestines harvested. Urine was collected for determination of prostanoids. Indomethacin was measured in urine, feces, and in the luminal contents and mucosal tissues of the proximal, middle, and distal regions of SI and LI, as well as from the tip of cecum.

In chronic indomethacin treatment study, mice were given control diet or indomethacin diet (20 ppm, Harlan, Madison, WI, USA) for 7 days. Fecal pellets were collected before and after treatment from each individual. Urine samples were collected for the determination of prostanoids on day 7 and mice were sacrificed on day 8 for tissue sampling along the GI tract.

## Study 2: Pharmacokinetics of indomethacin in antibiotics-treated mice

Individually housed mice were treated with or without the antibiotic cocktail for 5 days, with free access to a regular chow diet. Water for both treatment groups was spiked with aspartame. Body weight, food intake, and water intake of each mouse were followed daily throughout the study. 10 mg/kg indomethacin (in PEG400) was administered to mice in both antibiotic-treated and control groups at 12 pm. For the evaluation of indomethacin pharmacokinetics, blood were sampled from mouse tail vein at 1, 2, 4, 6, 8, 24, 30, 48 hr post indomethacin administration. Plasma were collected and stored at -80°C for the measurement of indomethacin using liquid chromatography/mass spectrometry (LC/MS). For the evaluation of glucuronidation, urine and fecal pellets were collected with the use of metabolic cages at 4, 8, 12, and 24 hr post indomethacin administration. Samples were stored at −80°C for the measurement of indomethacin and its metabolites using LC/MS.

## DNA extraction for microbiota composition analysis

Bacterial DNA was isolated from samples (fecal pellets, luminal contents, and mucosal tissues) using PSP Spin Stool DNA Plus Kit (Stratec, Berlin, Germany) with a slight modification. Briefly, samples were thawed on ice and transferred to Lysing Matrix E tubes (MP Biomedicals, Solon, OH, USA) with 1400 µl of stool stabilizer from the PSP kit. They were then disrupted using the TissueLyser II (Qiagen, Valencia, CA, USA) for 6 min at 30 Hz. Samples were then heated at 95°C for 15 min, cooled on ice for 1 min, and spun down at 13,400 $g$ for 1 min. The supernatant was then transferred to the PSP InviAdsorb tubes and the rest of the protocol for the PSP Spin Stool DNA Plus was followed according to the manufacturer's instructions. To maximize the extraction efficiency, each sample underwent two rounds of elution. Extracted DNA was quantified using NanoDrop 1000 (Thermo Scientific, Wilmington, DE, USA) and stored at −20°C for future use.

Every DNA extraction included a negative extraction control in which water was used instead of fecal pellets. All controls went through the same DNA extraction process as well as following amplification and sequencing processes.

## 16S rRNA gene quantification

Quantification of 16S rRNA gene was performed by real-time PCR using the Taqman method in triplicate reactions with 10 ng of DNA per reaction. Degenerate bacterial 16S rRNA gene-specific primers were used for amplification and their sequences were as follows: forward primer, 5'-AGAGTTTGATCCTGGCTCAG-3; reverse primer, 5'-CTGCTGCCTYCCGTA-3'; probe: 5' - /56-FAM/ TAA +CA+C ATG +CA+A GT+C GA/3BHQ_1/ - 3'; + precedes the position of LNA base.

Quantitative PCR was performed on a 7900HT Real-Time PCR System (Applied Biosystems, Grand Island, NY, USA). Thermocycling was performed as follows: initiation at 95°C for 5 min followed by 40 cycles of 94°C × 30 s, 50°C × 30 s, and 72°C × 30 s. Signals were collected during the elongation step at 72°C.

A standard curve prepared from a near full length clone of *Escherichi coli* 16S inserted into a Topo Vector was used for normalization for each run of real-time PCR.

## V1-V2 16S rRNA gene region amplification and sequencing

The V1-V2 region has performed well in reconstruction experiments and been used extensively previously in studies of the intestinal microbiome (*Liu et al., 2007*; *Chakravorty et al., 2007*; *Lozupone et al., 2007*; *Turnbaugh et al., 2009*; *Wu et al., 2010*; *Song et al., 2013*), and so was chosen here. A total of 100 ng of DNA was amplified with barcoded primers annealing to the V1-V2 region of the 16S rRNA gene using AccuPrime Taq DNA Polymerase System with Buffer 2 (Life Technologies, Grand Island, NY, USA). PCR reactions were performed on a thermocycler using the following conditions: initiation at 95°C for 5 min followed by 20 cycles of 95°C × 30 s, 56°C × 30 s, and 72°C × 1 min 30 s, then a final extension step at 72°C for 8 min. The amplicons from each DNA sample, which was amplified in quadruplicate, were pooled and purified with Agencourt AMPure XP

beads (Beckman Coulter, Beverly, MA, USA) following the manufacturer's instructions. Purified amplicon DNA samples were then sequenced using the 454/Roche GS FLX Titanium chemistry (454 Life Sciences, Branford, CT, USA). All novel sequence data were deposited at NCBI's Sequence Read Archive under Accession Numbers SRP 059293 and SRP 068846.

## 16S rRNA gene sequencing analysis and bioinformatics

Sequence data were processed with QIIME v 1.8.0 (*Caporaso et al., 2010b*) using default parameters. Firstly, samples with less than 200 counts were removed from further analysis. Sequences were clustered into operational taxonomic units (OTUs) at 97% similarity and then assigned taxonomy using the uclust consensus taxonomy classifier. Sequences were aligned using PyNAST (*Caporaso et al., 2010a*) and a phylogenetic tree was constructed using FastTree (*Price et al., 2009*). Weighted and unweighted UniFrac (*Lozupone and Knight, 2005*) distances were calculated for each pair of samples for the assessment of community similarity and generation of principal coordinate analysis (PCoA) plots. Taxonomic composition and alpha diversity were generated for each sample. To compare bacterial abundances across sample groups, *group_significance.py* was used with default parameters. To estimate the functional profile for each microbiota sample, the reads were analyzed with Phylogenetic Investigation of Communities by Reconstruction of Unobserved States (PICRUSt) version 1.0.0 (*Langille et al., 2013*) following the instructions. Predicted metagenomes were collapsed into KEGG (Kyoto Encyclopedia of Genes and Genomes) pathways (*Ogata et al., 1999*) and analyzed with STAMP (*Parks and Beiko, 2010*).

## Pharmacokinetic analysis

Plasma indomethacin concentrations at 1, 2, 4, 6, 8, 24, 30, 48 hr post administration were plotted against time to generate the 'plasma indomethacin concentration versus time curve'. With this curve, the area under the curve ($AUC_{total}$) was calculated according to the trapezoidal rule and the elimination rate constant ($K_{el}$) was obtained as the slope value. The half-life ($t_{1/2}$) was calculated as $t_{1/2} = ln2/k_{el}$. The apparent volume of distribution $V_d$ was calculated as $V_d = dose/C_0$. $C_0$ was extrapolated using the plasma drug concentration versus time curve. The oral clearance Cl was calculated as $Cl = dose/AUC_{total}$.

## Histological analysis of small intestinal damages

Histology of the injured SI was analyzed as described (*Imaoka et al., 2010*). Briefly, SIs were removed and perfused with phosphate buffered saline (PBS). Tissues were opened along the antimesenteric attachment and pinned down for macroscopic examination. The injured segments of the small intestine were trimmed, fixed overnight in 4% (vol/vol) paraformaldehyde at 4°C, washed with PBS, and dehydrated with ethanol before embedding in paraffin. The sections were cut and stained with hematoxylin and eosin (H&E) staining.

## Sample preparation for mass spectrometric analysis

### Prostanoids

Mouse urine (~100 µl) was spiked with 50 µl mixed stable isotope labelled internal standards. The sample was derivatized with 75 µl methoxime (in HCl) for 15 min at room temperature before solid-phase extraction (SPE).

### Indomethacin

Plasma samples (~10 µl) were mixed with 50 µl indomethacin internal standard (300 ng in ACN), 20 µl formic acid and 900 µl $H_2O$. The samples were vortexed and centrifuged before solid phase extraction (SPE). Tissue and fecal samples were homogenized in 1 ml Millipore $H_2O$ and briefly centrifuged. The samples were added with 2500 ng d4-indomethacin, vortexed, and incubated at room temperature for 15 min. The samples were centrifuged at 16,000 *g* for 15 min and the upper layer was transferred to a new tube. 100 µl of the supernatant was added with 900 µl $H_2O$ before solid phase extraction (SPE).

## Indomethacin metabolites

Urine samples (20 µl) were mixed with 40 µl indomethacin internal standard (40 ng in ACN), 400 µl sodium acetate and 10 µl β-Glucuronidase from Helix pomatia. After hydrolysis at 37°C for 4 hr, the samples were mixed with 20 µl formic acid and 500 µl $H_2O$. Another urine sample (20 µl) was mixed with 40 µl indomethacin internal standard (40 ng in ACN), 20 µl formic acid and 900 µl $H_2O$ without hydrolysis treatment. Dry stool samples were weighed before extraction with 1.7 mL of sodium acetate (0.2M, pH=5.0) using stainless steel-beads and a TissueLyser homogenizer (Qiagen, Valencia, CA, USA). The supernatant after centrifugation was divided to two aliquots. One aliquot was mixed with 40 µl indomethacin internal standard (40 ng in ACN), 20 µl formic acid and 300 µl $H_2O$ before SPE. The other aliquot was mixed with 40 µl indomethacin internal standard (40 ng in ACN) and 15 µl β-Glucuronidase. The β-Glucuronidase hydrolysis was performed at 37°C for 4 hr. The samples after hydrolysis were then mixed with 20 µl of formic acid and 300 µl $H_2O$ before SPE.

## Solid-phase extraction

SPE was performed according to the Manufacturer's instructions (Strata-X, Phenomenex, Torrance, CA, USA). Indomethacin and its metabolites were eluted with 1 ml methanol. Prostanoid metabolites were eluted with 1 ml ethyl acetate with 5% ACN.

## Calibration curves for mass spectrometric analysis

To calculate the precise relative amount of indomethacin metabolite, standard curves were prepared in mouse urine for indomethacin, and acyl-β-D-glucuronide Indomethacin (Santa Cruz Biotechnology, Dallas, TX, USA). Individual stock solutions of each compound (100 ng/µl) were prepared in ACN and stored at −80°C. Working solutions were prepared by mixing equal amounts of corresponding stock solutions and performing serial dilutions with ACN. Seven-point calibration samples (0, 0.032, 0.16, 0.8, 4, 20 and 100 ng/µl) for indomethacin and its metabolite were prepared. One large urine sample was obtained from mice without exposure to indomethacin. For each sample, 40 µl of (1 ng/µl) d4-Indomethacin (Santa Cruz Biotechnology, Dallas, TexasTX, USA), 10 µl calibration standards were added to 20 mouse urine. The calibration curves were also prepared with β-Glucuronidase hydrolysis. The samples were extracted by SPE before LC/MS.

## Liquid chromatography/mass spectrometry

Indomethacin and its metabolites were measured using a TSQ Quantum Ultra triple quadrupole mass spectrometer (Thermo Scientific, Wilmington, DE, USA) equipped with an ESI ion source. The Mass Spectrometer was connected to a Thermo Scientific Accela HPLC Systems and equipped with a PAL auto sampler and thermocontroller (set at 4°C). The CSH C18 Column (2.1 mm Xx 150 mm, 130Å, 1.7 µm, Waters) was used at a constant 40°C. The mobile phase (A) (90% $H_2O$/10% (B), 0.2% acetic acid) and mobile (B) (90% ACN/10% methanol) was used at a flow rate of 350 µl/min with a binary gradient (0–12 min, 10–50% B; 12–12.5 min, 50–100% B; 12.5–16 min, 100% B; 16.2–20 min, 10% B). Mass spectrometry was performed in negative mode. The transition for Indomethacin and d4-Indomethacin are 355.9>311.9 and 359.9>315.9, respectively. The transition for acyl-β-D-glucuronide Indomethacin is 533.1>193.3. Both Q1 and Q3 were operated at 0.7 m/z FWHM. Peak area ratios of target analytes to d4-Indomethacin internal standards were calculated by Xcalibur Quan software. The data were fitted to the calibration curves to calculate the precise relative amount of indomethacin metabolites.

Prostanoid metabolites were measured using a Waters Acquity UPLC system comprising a binary pump, an autosampler, and a Xevo TQ-S triple quadrupole mass spectrometer equipped with an electrospray ionization source (Waters, Milford, MA, USA). Chromatographic separation was performed on a Waters UPLC CSH C18 column (2.1 mm x 150 mm, 130 Å, 1.7 µm). The UPLC mobile phases consisted of (A) (95%$H_2O$/5% (B), pH=5.7) and (B) 95%ACN/5% methanol. The initial gradient began with 0% B. Mobile phase B increased linearly to 10% at 17 min, to 10.5% at 17.5 min, to 11.5% at 32 min, to 20% at 35 min, to 43% at 43 min, to 100% at 43.5 min, and finally go back to 0% at 45.5 min. A 0.35 ml/min flow rate was used throughout the UPLC gradient. The autosampler temperature was set at 4°C and the UPLC column was heated at 50°C. The MS was operated under negative ion mode at MRM mode. The transitions were monitored as previously described (*Song et al., 2007*). Briefly, systemic production of $PGI_2$, $PGE_2$, $PGD_2$, and $TxB_2$ was determined by

quantifying their major urinary metabolites: 2, 3-dinor 6-keto $PGF_{1\alpha}$ (PGI-M); 7-hydroxy-5, 11-diketo-tetranorprostane-1, 16-dioic acid (PGE-M); 11, 15-dioxo-$9_\alpha$-hydroxy-2, 3, 4, 5-tetranorprostan-1, 20-dioic acid (tetranor PGD-M); and 2, 3-dinor $TxB_2$ (Tx-M), respectively. esults were normalized with creatinine (Oxford Biomedical Research, Rochester Hills, MI, USA). Peak areas were obtained using MassLynx software (Waters).

## Statistical analysis

Statistical analyses were performed using PRISM or QIIME (*Caporaso et al., 2010b*). Mann-Whitney test, Wilcoxon test, multiple t test, or QIIME analysis were conducted as indicated in figure legend. All data were expressed as means ± SEM.

## Acknowledgements

We gratefully acknowledge the technical support and advice of Christian Hoffmann, Stephanie Grunberg, Aubrey Bailey, Alice Laughlin, Helen Zou, and Wenxuan Li-Feng. Dr. FitzGerald is the Robert L. McNeil, Jr. Professor in Translational Medicine and Therapeutics. Supported by HL 117798 from NHLBI.

## Additional information

### Funding

| Funder | Grant reference number | Author |
| --- | --- | --- |
| National Heart, Lung, and Blood Institute | 1U54HL117798 | Xue Liang<br>Xuanwen Li<br>Garret A FitzGerald |

The funders had no role in study design, data collection and interpretation, or the decision to submit the work for publication.

### Author contributions

XLi, Conception and design, Acquisition of data, Analysis and interpretation of data, Drafting or revising the article; KB, FDB, GAF, Conception and design, Analysis and interpretation of data, Drafting or revising the article; XLi, Acquisition of data, Drafting or revising the article; DRA, Conception and design, Drafting or revising the article

### Author ORCIDs

Frederic D Bushman, http://orcid.org/0000-0003-4740-4056

### Ethics

Animal experimentation: This study was performed in strict accordance with the recommendations in the Guide for the Care and Use of Laboratory Animals of the National Institutes of Health. All of the animals were handled according to approved institutional animal care and use committee (IACUC) protocols (#803903) of the University of Pennsylvania. Throughout the study, every effort was made to minimize suffering.

## Additional files

### Major datasets

The following datasets were generated:

| Author(s) | Year | Dataset title | Dataset URL | Database, license, and accessibility information |
|---|---|---|---|---|
| Xue Liang, Frederic D Bushman, Garret A FitzGerald | 2015 | Sequence Read Archive | http://www.ncbi.nlm.nih.gov/sra/?term= SRP059293 | Publicly available at the NCBI Sequence Read Archive (Accession no. SRP059293). |
| Xue Liang, Frederic D Bushman, Garret A FitzGerald | 2015 | Sequence Read Archive | http://www.ncbi.nlm.nih.gov/sra/?term= SRP068846 | Publicly available at the NCBI Sequence Read Archive (Accession no. SRP068846). |

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
