## [Decision Letter]

Thank you for choosing to send your work entitled "Bidirectional interactions between indomethacin and the murine intestinal microbiota" for consideration at *eLife*. Your submission has been reviewed by Fiona Watt (Senior Editor) and three reviewers, one of whom is a member of our Board of Reviewing Editors.

Summary:

Investigations of the effects of NSAIDs, such as indomethacin on the gut microbiota, and the role of the microbiota in drug pharmacokinetics are significant and impactful questions for studies of the microbiome. While all three reviewers found the topic of your manuscript interesting and innovative, they all expressed major concerns about the findings and their interpretation and contextualization. Furthermore, the reviewers expressed that substantial revision would be required for reconsideration.

Essential revisions:

Major concerns reflected in the reviewers' comments below spanned the dosage of indomethacin and the selection of study time points, the 16S analysis and its findings given the large effect of the vehicle, and the underexploration experimentally and discussion regarding how a drug that mediates SI injury results in broader biogeographical shifts, e.g. LI microbiome.

Reviewer #1:

While the field and topic are appropriate for *eLife* and represent a strong effort to coordinate biogeographical data with reciprocal effects between the microbiota and the host (a field that is becoming a scientific topic of increasing), there are several concerns regarding the methodology and interpretation of this work.

The major concerns relate to experimental design [the dosage of indomethacin and the time points selected in this study (6h) that are used to describe the work as longitudinal]. Please justify the choice to use of such a traumatic dosage, rather than a dosage, more relevant to the use of the drug for therapeutic/symptom management purposes given the goal of the study to use mouse models to mechanistically unravel host-microbe-drug interaction in humans.

Further, given the changes that occur in the microbiome over dietary, circadian cycles, etc, a longer timeframe would provide more relevant information as to the effects of indomethacin on the microbiota. Ideally, the experiments in this study would be performed over a time-period of weeks with sub-inflammatory dosages of indomethacin and include such temporal data.

Specific points:

1) Please explain/consider choice of V1-V2 for 16S rRNA gene survey sequencing over V3-V4 as per Earth Microbiome project standards.

2) Add percentage variation explained to PCoA figure.

3) Results, paragraph two seem unclear/unsupported by data in the proximal LI.

4) Results, paragraph six: Confirmation by qPCR or another method like FISH would strengthen this section.

5) Given that the damage occurs in the SI, more weight should be given to the effect on the LI microbiota in the Discussion. Why might these changes occur here? Are there testable hypotheses that can be generated?

6) Are the fecal data in Figure 2 the same as the data used in the 'longitudinal' study in Figure 3? Please clarify/combine to avoid repetition of data.

7) In the 'longitudinal' studies (Figure 3), it almost seems more like the indomethacin treated mice were 'abnormal' at t=0 rather than differing from untreated controls at t=6h. Please clarify the selection of data to compare for statistical purposes.

Reviewer #2:

Liang and colleagues present a manuscript rich with data, but unfortunately, lacking a primer. While there is a wealth of information contained within this manuscript, it lacks scholarly context, a disservice to the quantity of data contained therein. The reader would be well served if the authors would provide a context for each experiment carried out. Moreover, the text contains many outdated citations and contains many typos, which would benefit from editing. Most importantly, this manuscript contains a lack of synthesis and interpretation of the data.

Specific criticism:

1) The Introduction contains citations from as far back as 1977, and neglects to cite contemporary work such as that by Boelsterli and colleagues. More contemporary citations should be referenced in both Introduction and Discussion.

2) The authors should include a brief description of jargon such as "Shannon index" etc. which are familiar to those within the field only, and not a broad audience.

3) What are the clinical implications of these findings? This should be addressed in the Discussion section. Does concurrent treatment with antibiotics impact the efficacy or the side effects of (chronic or acute) NSAID treatment? A literature review may answer this question, which should be incorporation within the Discussion.

4) In general, NSAID-mediated injury is observed in the small intestine. What is the significance of finding an expansion of *Erysipelotrichaceae* in the large intestinal mucosa?

5) Figure 3 has some issues: the authors mention separate clusters forming in the indomethacin group however they are not apparent in the figure, which looks like fully overlapping blue and black circles. Similarly, what is the significance of the two main clusters observed in the PEG-400 treated group? The black and blue points seem to be overlapping in all instances. Please clarify.

6) Can the authors comment on why indomethacin would upregulate the *Ruminococceae* spp? In a similar vein, the authors could comment on why PEG-400 may upregulate other species.

7) Supplemental figures: some are missing labels (e.g the histology image lacks any captions).

8) The authors should spend more time describing Figure 4, which contains very important information. The authors have "addressed the hypothesis" but not actually made a conclusion, nor correlated with how this impacts treatment regimens with NSAIDs (see comment #3). An allusion is made towards this in paragraph four of the Discussion, but a deeper discussion is warranted.

9) The authors find that perturbations in intestinal microbiota specifically occur in cecum, large intestine and feces; however, the bulk of assault by NSAIDs occurs in the small intestine, as clearly demonstrated by at least three papers by Boelsterli and colleagues on NSAID damage and its alleviation. How do compositional changes in the more distal portion of the GI tract impact pathology more proximally?

10) What is the substance being measured in Figure 5—figure supplement 1?

11) The authors note a depletion of 16S copy numbers and decrease in microbial diversity following antibiotic treatment, and an expansion of Proteobacteria. Do Proteobacteria impact GI toxicity of indomethacin and if so, how?

12) The suggestion that bacteria contain β-glucuronidases, along with one citation from Roberts et al. is not sufficient to connect this work with the well established data by Redinbo and coworkers showing not only that bacteria have β-glucuronidases, but that those enzymes are directly involved in NSAID-induced GI damage. Furthermore, they show that inhibiting those bacterial enzymes can reduce this damage.

Reviewer #3

Liang et al. present a series of studies testing the impact of the NSAID drug indomethacin on the gut microbiota as well as the role of the microbiota in drug pharmacokinetics. While I have concerns about the first claim, the latter finding is quite convincing and would be a nice addition to the literature. The major strengths of this paper are the large sample size, both in terms of animals and locations within the gut, and the pharmacokinetic analyses. The manuscript is clearly written and the figures are easy to read.

1) The 16S analyses could be improved. As is, I'm not convinced that the drug changed the gut microbiota much more than the vehicle (which seems to have had a large effect).

Figure 2 show some differences in α-diversity, but they are either inconsistent between lumen vs. tissue samples, inconsistent between observed OTUs and the Shannon index (both of which measure species richness, so they should match), small magnitude, or restricted to a small number of gut locations.

Figure 2 is intriguing, but it's unclear how this taxon was found. Were the FDR values corrected for all taxa at this level or just for the different locations?

Figure 2 may be significant relative to vehicle but is not significantly different from the untreated animals, raising questions as to its biological relevance.

In Figure 3 the authors state that the indomethacin treated animals cluster by timepoint but is not supported by the current color labels. The only clear clustering is for the vehicle but this doesn't seem to group by timepoint (there's no mention of the reason – could this be due to housing?). It's also unclear why the groups were all presented separately, instead of testing if the drug changed the gut microbiota relative to the control treatments and baseline samples.

In Figure 3 the vehicle/drug groups are slightly lower before treatment – what is the explanation for this?

Figure 3 is used to conclude that many taxa are significantly affected by the drug in fecal samples, raising the question as to why these weren't found in the previous analysis of colon contents. The same concern about FDR correction (see 2D above) applies here. Also, nearly all of these taxa are similar in the 3 treatment groups after treatment – the significance appears to be due to a difference in drug treated group at the baseline timepoint before treatment. While I appreciate the idea of using each mouse as its own control, the consistency of these differences at the baseline timepoint causes me to worry about what might have been different between the mice assigned to each treatment group.

A more general issue is that many of the 16S comparisons are qualitative and lack proper statistical tests. For example, Results, paragraph two includes many statements that need to be backed up by statistics. It would also help to include numbers in the text to give the reader a sense of the magnitude of each change and the variance within each group.

2) The data in Figure 4 is excellent and Figure 5 is particularly striking. Together, they show the importance of the gut microbiota in the pharmacokinetics of indomethacin. However, I have some concerns as to the novelty of this finding. As the authors mention in the Results section, Saitta et al., 2014 already conclusively showed that bacterial β-glucuronidase contributes to the GI damage caused by indomethacin. While it is very comforting to confirm that this is due to deconjugation its not that big a step forward.

A bigger advance would be to support the authors' hypothesis that changes in the gut microbiota following drug treatment then alters drug levels or side effects. This might be done by doing transplantation experiments into germ-free mice from donors that were treated with indomethacin or vehicle controls followed by drug treatment. Better yet, maybe some of the putative drug-associated taxa that the authors have identified could be used to colonize germ-free or antibiotic treated mice prior to indomethacin treatment. It would also be possible to quantify the abundance of β-glucuronidase pre- and post-treatment by quantitative PCR, enzymatic assays, or metagenomic shotgun sequencing.

3) I'm not sure what the "tissue" microbiota represents as microbes are typically found in the lumen and outer mucosal layer with very few penetrating the tissue in wild-type animals. More information is needed in the Materials and methods section to explain the sample collection procedure. How was the lumen washed away? Was the mucus retained prior to homogenization?

---

## [Author Response]

*Essential revisions:*

Major concerns reflected in the reviewers' comments below spanned the dosage of indomethacin and the selection of study time points, the 16S analysis and its findings given the large effect of the vehicle, and the underexploration experimentally and discussion regarding how a drug that mediates SI injury results in broader biogeographical shifts, e.g. LI microbiome.

In response to essential revisions from the editor:

1) The dose of indomethacin was selected for the following reasons:

According to FDA guidance, a dose of 10mg/Kg body weight in mice is equivalent to 0.81mg/Kg body weight in humans (48.8 mg for a 60 Kg human). This is a dosage that is commonly used for treating acute pain as well as rheumatoid arthritis and osteoarthritis. This dose is also widely used in experimental animal models to induce intestinal damage (Fukumoto et al. 2011, Tanigawa et al. 2013).

At this dose, indomethacin successfully inhibits both COX-1 and COX-2 enzymes in vivo, as indicated by the significantly reduced urinary prostaglandin metabolite levels. This information was provided in Figure 2—figure supplement 2. This dose has been shown in our study as well as in previous reports to induce small intestinal lesions. Our data are provided in Figure 2—figure supplement 1.

In our chronic treatment experiment, 20 ppm of indomethacin was provided in the diet. For an adult mouse weighing 25g and eating 4g per day, this amounts to 3.2mg/Kg/day. This dose was chosen based on previous reports showing suppression of prostanoids by indomethacin (Chiu et al. 2000, Leibowitz et al. 2014, Fjaere et al. 2014). This dose equivalent is less than the acute dosing regimen to ensure tolerability and regulatory compliance. The reasons for the doses chosen are now specified more clearly in the revised manuscript.

2) The 6-hour treatment was selected for the following reasons:

Six hours is approximately the half-life of indomethacin in rodents. Six hours post dose is before the appearance of indomethacin induced lesions; therefore it allows us to study the potential role of gut microbiota in the development of indomethacin-induced intestinal damage. The lesions in the small intestine were detected at 24 hours. Hence, analysis at 6 hours post treatment excluded potential lesion-induced changes in microbial composition. These reasons are now spelled out more clearly in the revised manuscript.

3) 16S rRNA gene sequencing—effects of vehicle and choice of the V1-V2 window:

The vehicle used in the acute studies did have a notable effect. PEG was needed to solubilize indomethacin for the acute dosing studies, but PEG is also a component of lavage mixtures used for bowel preps in human patients and so strongly affects the microbes present. Here the PEG treatment resulted in softer pellets and altered microbiota. Use of PEG alone as a control indicates an impact of indomethacin over and above the vehicle effect in the acute dosing study. In the long-term dosing study, PEG was not used and some of the same bacterial compositional changes were observed despite lower systemic indomethacin exposure, again consistent with both PEG and indomethacin affecting microbial composition.

Regarding the choice of the V1-V2 region of the 16S rRNA gene for sequencing, this region has been found to yield good partitioning in sample sets from many different control studies and experiments in the laboratories of Gordon, Knight, Bushman and others (Liu et al. 2007, Chakravorty et al. 2007, Wu et al. 2010). All 16S sequence windows used for analysis recover reads from some bacteria better than others. For example, the V1-V2 window is known to be biased against Bifidobacteria, but this group is not commonly seen in the mouse fecal microbiome. We thus feel that the V1-V2 region is a reasonable choice here. We have added discussion of these points to the revised manuscript.

4) The link between small intestinal injury and microbial shifts in the large intestine:

Although we didn’t reveal statistically significant changes in bacterial communities residing in the small intestine (SI), this doesn’t necessarily allow us to conclude that indomethacin is not influencing the SI microbiota. As indicated in Figure 1, SI microbiota express greater variability among individuals than in the large intestine (LI), which may mask indomethacin-induced compositional changes.

Indomethacin also causes damage in the lower GI tract. Significantly increased large intestinal permeability (Suenaert et al. 2003), colonic ulceration and bleeding (Oren, Ligumsky 1994), multiple colonic perforations (Loh et al. 2011) and hemorrhage (Langman et al. 1985) have been reported in patients receiving indomethacin. Evidence suggests that incidence of upper and lower GI complications induced by NSAIDs are similar (Sostres et al. 2013). Compositional changes in the large intestine induced by indomethacin may be involved in these inflammatory processes as well.

The intestinal bacteria are constantly interacting with other species and with the host, in part through the metabolites they synthesize. Altered bacterial populations may secrete molecules that modify the local environment or enter the circulation, hence changing inter-bacterial interactions or host physiology, possibly influencing pathogenesis of intestinal lesions. Possible mechanisms are the topic of ongoing study – whether microbial populations are involved mechanistically in intestinal injury is unclear. We now spell this out in the revised manuscript.

Reviewer #1:

*While the field and topic are appropriate for eLife and represent a strong effort to coordinate biogeographical data with reciprocal effects between the microbiota and the host (a field that is becoming a scientific topic of increasing), there are several concerns regarding the methodology and interpretation of this work. The major concerns relate to experimental design [the dosage of indomethacin and the time points selected in this study (6h) that are used to describe the work as longitudinal]. Please justify the choice to use of such a traumatic dosage, rather than a dosage, more relevant to the use of the drug for therapeutic/symptom management purposes given the goal of the study to use mouse models to mechanistically unravel host-microbe-drug interaction in humans. Further, given the changes that occur in the microbiome over dietary, circadian cycles, etc, a longer timeframe would provide more relevant information as to the effects of indomethacin on the microbiota. Ideally, the experiments in this study would be performed over a time-period of weeks with sub-inflammatory dosages of indomethacin and include such temporal data.*

We appreciate the reviewer’s comments on the experimental design, including dose selection, time point selection, and rationale for using V1-V2 region of 16S rRNA gene. These points are addressed above.

We agree that the influence of long-term indomethacin treatment on microbial composition is an important issue, so following the reviewer’s comment we conducted an additional experiment where mice received a control diet or indomethacin in the diet (20 ppm) for 7 days. The data and interpretation are presented in the revised manuscript. Chronic treatment also induced the expansion of *Peptococcaceae* and *Erysipelotrichaceae* in the GI tract and *Ruminococcus* and *Anearoplasma* in the feces, consistent with previous results from the study of acute treatment. Evidently these genera responded to indomethacin both in the initial phase and in the longer term. We also observed that several acute treatment-induced changes were less significant in the long-term dosing study. It could be that these changes are involved in the initial response only, or that the temporal variation overrides the drug-induced changes. Additional possibilities include the differing magnitude of systemic drug exposure, the methods of drug delivery, and non-identical microbiota in mice at the start of the experiment. Further studies, outside the scope of this manuscript, will address these possibilities.

*Specific points: 1) Please explain/consider choice of V1-V2 for 16S rRNA gene survey sequencing over V3-V4 as per Earth Microbiome project standards.*

The use of the V1-V2 16S rRNA gene for sequencing is justified under Essential Revisions comment #3 above.

*2) Add percentage variation explained to PCoA figure.*

The percentage variation has been added to the axes on each PCoA plot as requested.

*3) Results, paragraph two seem unclear/unsupported by data in the proximal LI.*

These are now supported with numerical data.

*4) Results, paragraph six: Confirmation by qPCR or another method like FISH would strengthen this section.*

We agree with the reviewer that changes in bacterial abundance would be strengthened by conformation with other methods. As mentioned above, we have replicated key compositional changes seen in the acute study in the newly added long term dosing study.

*5) Given that the damage occurs in the SI, more weight should be given to the effect on the LI microbiota in the Discussion. Why might these changes occur here? Are there testable hypotheses that can be generated?*

Discussion on microbial shifts in LI and damage in SI are provided under Essential Revisions comment #4 above as well as in the revised Discussion.

6) Are the fecal data in Figure 2 the same as the data used in the 'longitudinal' study in Figure 3? Please clarify/combine to avoid repetition of data.

The fecal data presented in Figure 2 and Figure 3 are overlapping but are not repeats. Figure 2 illustrates the geographic changes caused by indomethacin treatment after 6 hours across three groups. The fecal microbiota was considered as an extension of the luminal intestinal microbiota, hence provided as part of this analysis. However, since this is cross-sectional analysis, only 6h fecal data are presented in Figure 2. Figure 3, on the other hand, illustrates the temporal changes in fecal microbiota within each individual mouse by comparing microbial composition prior to (0h) and post (6h) indomethacin treatment in individual groups. Therefore, the fecal microbiota data were not repeats.

*7) In the 'longitudinal' studies (Figure 3), it almost seems more like the indomethacin treated mice were 'abnormal' at t=0 rather than differing from untreated controls at t=6h. Please clarify the selection of data to compare for statistical purposes.*

We agree with the reviewer that at t = 0h, there were differences between treated mice and control mice. It is well known that there is large inter-individual variation in intestinal microbiota composition. We randomized allocation of the mice to each group and importantly controlled for cage effects as a part of the statistical analysis. We are comparing within-individual changes due to indomethacin changes, so each individual serves as their own control, lending more support to our conclusion.

Reviewer #2:

*Liang and colleagues present a manuscript rich with data, but unfortunately, lacking a primer. While there is a wealth of information contained within this manuscript, it lacks scholarly context, a disservice to the quantity of data contained therein. The reader would be well served if the authors would provide a context for each experiment carried out. Moreover, the text contains many outdated citations and contains many typos, which would benefit from editing. Most importantly, this manuscript contains a glaring lack of synthesis and interpretation of the data, which is the main reason this reviewer is not keen on accepting the manuscript as is.*

We appreciated the reviewer’s comments on the language and citations, and have revised the manuscript extensively in response.

*Specific criticism: 1) The Introduction contains citations from as far back as 1977, and neglects to cite contemporary work such as that by Boelsterli and colleagues. More contemporary citations should be referenced in both Introduction and Discussion.*

In the revised manuscript we’ve cited the work of Boelsterli and others in the Introduction and Discussion.

*2) The authors should include a brief description of jargon such as "Shannon index" etc. which are familiar to those within the field only, and not a broad audience.*

We have added a brief description of the Shannon Index as requested.

*3) What are the clinical implications of these findings? This should be addressed in the* Discussion section*. Does concurrent treatment with antibiotics impact the efficacy or the side effects of (chronic or acute) NSAID treatment? A literature review may answer this question, which should be incorporation within the Discussion.*

Clinical implications are now discussed more fully in the revised Discussion.

*4) In general, NSAID-mediated injury is observed in the small intestine. What is the significance of finding an expansion of Erysipelotrichaceae in the large intestinal mucosa?*

We discussed the issue that changes were detected in colonic microbiota while damage was present in the small intestine in response to the editor under Essential Revisions comment #4 as well as in the revised Discussion. Expansion of *Erysipelotrichaceae* has been associated with parenteral nutrition induced liver injury, obesity, colorectal cancer, and Crohn’s disease, indicating a pro-inflammatory role of *Erysipelotrichaceae*. However, due to current lack of knowledge about these bacteria and their potential causality in these diseases, it is too early to conclude the significance of *Erysipelotrichaceae* expansion in indomethacin-induced GI toxicity.

*5) Figure 3 has some issues: the authors mention separate clusters forming in the indomethacin group however they are not apparent in the figure, which looks like fully overlapping blue and black circles. Similarly, what is the significance of the two main clusters observed in the PEG-400 treated group? The black and blue points seem to be overlapping in all instances. Please clarify.*

In Figure 3, we performed ADONIS test (Anderson 2001) over the weighted UniFrac distances to test the clustering of samples based on treatment, and the results were reported in the Results section.

*6) Can the authors comment on why indomethacin would upregulate the Ruminococceae spp? In a similar vein, the authors could comment on why PEG-400 may upregulate other species.*

The reviewer asked for a possible explanation of the expansion of *Ruminococceae spp* upon indomethacin treatment as well as similar effects due to PEG-400 treatment. We think there could be multiple mechanisms by which xenobiotics cause compositional changes in gut microbiota, including those previously reported by others. One example would be the cross-feeding model. Certain commensal bacteria can directly consume xenobiotics and secrete metabolites into the environment, which may be used by other bacteria to boost their growth. Another example could be a host-dependent influence, where xenobiotics affect host characteristics (especially in the local environment of the intestine) and induce changes in immune regulation of gut bacteria. Chassaing and colleagues (Chassaing et al. 2015) reported that dietary emulsifiers, such as carboxymethylcellulose and polysorbate-80, altered microbial localization as well as OTU abundances. However, they didn’t establish the causal relationship between the localization and altered OTUs. Further studies are needed to understand the mechanism of xenobiotic-induced changes in bacteria.

*7) Supplemental figures: some are missing labels (e.g the histology image lacks any captions).*

We have added captions for the histology images as requested.

*8) The authors should spend more time describing Figure 4, which contains very important information. The authors have "addressed the hypothesis" but not actually made a conclusion, nor correlated with how this impacts treatment regimens with NSAIDs (see comment #3). An allusion is made towards this in paragraph four of the Discussion, but a deeper discussion is warranted.*

We have added more discussion on Figure 4 in the Result and Discussion in the revised manuscript.

*9) The authors find that perturbations in intestinal microbiota specifically occur in cecum, large intestine and feces; however, the bulk of assault by NSAIDs occurs in the small intestine, as clearly demonstrated by at least three papers by Boelsterli and colleagues on NSAID damage and its alleviation. How do compositional changes in the more distal portion of the GI tract impact pathology more proximally?*

Discussion on microbial shifts in LI and damages in SI are provided under No.4 above as well as in the revised Discussion.

*10) What is the substance being measured in Figure 5—figure supplement 1?*

The substances being measured in Figure 5—figure supplement 1 are labeled in each panel: Acyl-β-D-glucuronide Indomethacin in top panel; indomethacin in middle panel; and d4-indomethacin in bottom panel.

*11) The authors note a depletion of 16S copy numbers and decrease in microbial diversity following antibiotic treatment, and an expansion of Proteobacteria. Do Proteobacteria impact GI toxicity of indomethacin and if so, how?*

The reviewer pointed out that the expansion of Proteobacteria following antibiotic treatment and questioned whether this would influence the GI toxicity of indomethacin. The antibiotic treatment model was set up to study the role of bacteria in indomethacin metabolism rather than GI toxicity, since the expansion of Proteobacteria was not caused by indomethacin treatment. Additionally, β – glucuronidase activity hasn’t been reported in Proteobacteria according to previous studies (Dabek et al. 2008, Gloux et al. 2011), hence it is not likely that these bacteria would influence the exposure of indomethacin in host. Therefore, current data do not support an influence of Proteobacteria in indomethacin-induced enteropathy.

*12) The suggestion that bacteria contain* β*-glucuronidases, along with one citation from Roberts* et al. *is not sufficient to connect this work with the well established data by Redinbo and coworkers showing not only that bacteria have* β

*-glucuronidases, but that those enzymes are directly involved in NSAID-induced GI damage. Furthermore, they show that inhibiting those bacterial enzymes can reduce this damage.*We thank the reviewer for bringing up previous work by Redinbo and coworkers, which is now cited. However, our study is the first to link pharmacokinetics and pharmacodynamics, providing a bidirectional impact between intestinal microbiota and indomethacin.

Reviewer #3:

*Liang* et al.

*present a series of studies testing the impact of the NSAID drug indomethacin on the gut microbiota as well as the role of the microbiota in drug pharmacokinetics. While I have concerns about the first claim (see below), the latter finding is quite convincing and would be a nice addition to the literature. The major strengths of this paper are the large sample size, both in terms of animals and locations within the gut, and the pharmacokinetic analyses. The manuscript is clearly written and the figures are easy to read. 1) The 16S analyses could be improved. As is, I'm not convinced that the drug changed the gut microbiota much more than the vehicle (which seems to have had a large effect).*

The reviewer was concerned about the effect of vehicle on the microbial composition. In our recently-conducted chronic treatment experiment, mice were given a control diet or indomethacin mixed in the diet (20 ppm) for 7 days—that is, the dosing was in the absence of the PEG carrier. The data and interpretation are presented in the revised manuscript. Briefly, the chronic treatment induced the expansion of *Peptococcaceae* and *Erysipelotrichaceae* in the GI tract and *Ruminococcus* and *Anearoplasma* in the feces, as in the acute treatment experiment, indicating these changes are independent of the vehicle effect. We also observed that several acute treatment-induced changes were less impressive in this chronic dosing study. This is addressed in the revised manuscript.

*Figure 2 show some differences in* α*-diversity, but they are either inconsistent between lumen vs. tissue samples, inconsistent between observed OTUs and the Shannon index (both of which measure species richness, so they should match), small magnitude, or restricted to a small number of gut locations.*

The reviewer pointed out that Figure 2 shows changes in α diversity along the GI tract that are different between luminal vs mucosal samples, and different between observed OTUs and Shannon index. The numbers of OTUs (richness) and diversity are not identical. Richness quantifies the number of types present, while diversity takes into account not just the number of types present but the evenness of distribution among individuals. The difference between luminal versus mucosal samples is not surprising and is consistent with published literature (Hill et al. 2010, Albenberg et al. 2014, Looft et al. 2014). The IBD literature emphasizes differences between luminal and mucosa microbiota (Carroll et al. 2011, Lavelle et al. 2015).

Figure 2 is intriguing, but it's unclear how this taxon was found. Were the FDR values corrected for all taxa at this level or just for the different locations?

In Figure 2, the statistics were carried out for each GI location. Comparisons were made using the *group_significance.py* script with Kruskal-Wallis tests in QIIME analysis. This script allows the comparison of taxa frequencies across sample groups, and returns a p value corrected by the Benjamini-Hochberg FDR procedure for multiple comparisons.

Figure 2 may be significant relative to vehicle but is not significantly different from the untreated animals, raising questions as to its biological relevance.

We agree with the reviewer that in Figure 2 indomethacin treatment was significant relative to vehicle. This still indicates a drug effect since indomethacin may counteract and reverse the vehicle effect. In our chronic treatment experiment, indomethacin induced expansion of *Peptococcaceae* and *Erysipelotrichaceae* in the GI tract significantly compared to control animals (Figure 2—figure supplement 5), indicating that these changes are independent of a vehicle effect.

In Figure 3 the authors state that the indomethacin treated animals cluster by timepoint but is not supported by the current color labels. The only clear clustering is for the vehicle but this doesn't seem to group by timepoint (there's no mention of the reason – could this be due to housing?). It's also unclear why the groups were all presented separately, instead of testing if the drug changed the gut microbiota relative to the control treatments and baseline samples.

In Figure 3, the previous version reported PCoA plots using unweighted UniFrac distance. The separation in the PEG400 group was due to a cage effect, which is why we carefully controlled for cage effects in our statistical analysis. In the new version, however, we decided to use the weighted UniFrac distances, since we are discussing the changes in relative abundances of bacteria. We also performed ADONIS test (Anderson 2001) and found significant clustering of before and after treatment samples. These results were reported in the revised Results.

In Figure 3 the vehicle/drug groups are slightly lower before treatment – what is the explanation for this?

The reviewer also highlighted the reduction in 16S rRNA gene copy numbers in vehicle/drug group in Figure 3. We think that this is a vehicle effect, which is likely to be attributable to its purgative effect. One possible explanation could be that by retaining water, an effect of PEG, the bacterial community was diluted, leaving fewer bacteria per unit weight in fecal samples. Further studies are needed to understand the exact mechanisms.

Figure 3 is used to conclude that many taxa are significantly affected by the drug in fecal samples, raising the question as to why these weren't found in the previous analysis of colon contents. The same concern about FDR correction (see 2D above) applies here. Also, nearly all of these taxa are similar in the 3 treatment groups after treatment – the significance appears to be due to a difference in drug treated group at the baseline timepoint before treatment. While I appreciate the idea of using each mouse as its own control, the consistency of these differences at the baseline timepoint causes me to worry about what might have been different between the mice assigned to each treatment group.

In Figure 3, comparisons and FDR corrections are conducted in the same way as for Figure 2. One of the reasons why these taxa were not found in analysis of colon contents is probably due to the difference between longitudinal analyses versus cross-sectional analyses. In fecal microbiota analysis, each mouse serves as its own control so that the experimental design was controlled for various confounding factors such as inter-mouse variability and caging effects. Hence, the real biological changes due to the drug treatment are less likely to be masked by other variables. In contrast, in the colon content analysis, comparisons were made between different groups of mice so that inter-mouse variability and caging effect may mask the influences caused by drug treatment. The reviewer also pointed out the differences among mice before treatment. We were aware of inter-mouse variability, so we carefully randomized animals into three groups and randomly conducted treatment on different days. All experiments were carried out by the same individual following the same procedure to minimize technical variables. Therefore, the baseline difference among mice reflects biological variance. We now discuss this in the revised manuscript.

A more general issue is that many of the 16S comparisons are qualitative and lack proper statistical tests. For example, Results, paragraph two includes many statements that need to be backed up by statistics. It would also help to include numbers in the text to give the reader a sense of the magnitude of each change and the variance within each group.

As requested, we now provide more numeric data and statistical results in the revised manuscript.

*2) The data in Figure 4 is excellent and Figure 5 is particularly striking. Together, they show the importance of the gut microbiota in the pharmacokinetics of indomethacin. However, I have some concerns as to the novelty of this finding. As the authors mention in the* Results section*, Saitta* et al.*, 2014 already conclusively showed that bacterial* β*-glucuronidase contributes to the GI damage caused by indomethacin. While it is very comforting to confirm that this is due to deconjugation its not that big a step forward.*

We agree that previous work has demonstrated the role of bacterial β-glucuronidase in GI damage by indomethacin. However, our study is the first to link pharmacokinetics and pharmacodynamics, providing a bi-directional impact between intestinal microbiota and indomethacin. Thus we highlight that bacteria may contribute to inter-individual variation in drug response among individuals. Given that the microbial composition undergoes circadian oscillation during the light-dark cycle, bacterial influence on pharmacokinetics may also explain intra-individual variation at various times of the day.

*A bigger advance would be to support the authors' hypothesis that changes in the gut microbiota following drug treatment then alters drug levels or side effects. This might be done by doing transplantation experiments into germ-free mice from donors that were treated with indomethacin or vehicle controls followed by drug treatment. Better yet, maybe some of the putative drug-associated taxa that the authors have identified could be used to colonize germ-free or antibiotic treated mice prior to indomethacin treatment. It would also be possible to quantify the abundance of* β*-glucuronidase pre- and post-treatment by quantitative PCR, enzymatic assays, or metagenomic shotgun sequencing.*

The reviewer pointed out that germ-free mouse would be a good model for further studies, including drug treatment and fecal transplantation experiments. We note that an advantage of using antibiotic treatments is that the mice studied will have developed normal gut and immune function by growth with normal colonization—gnotobiotic mice are known to be abnormal in these respects. Still, gnotobiotic studies would be a good addition. Unfortunately, during the study we did not have access to gnotobiotic mice. But these are on our list of future studies. In addition, we were especially interested to use antibiotics given the commonality of co-administration of NSAIDs and antibiotics, especially to orthopedic patients.

3) I'm not sure what the "tissue" microbiota represents as microbes are typically found in the lumen and outer mucosal layer with very few penetrating the tissue in wild-type animals. More information is needed in the Materials and methods section to explain the sample collection procedure. How was the lumen washed away? Was the mucus retained prior to homogenization?

We have explained the luminal and mucosal sample collection procedure more carefully in the revised Materials and methods session.